# Graph-Theoretic Insights into Bayesian Personalized Ranking for Recommendation

**Kai Zheng**[1,2,3]**, Jianxin Wang**[1,2,*]**, Jinhui Xu**[4,5,*]

[1]School of Computer Science and Engineering, Central South University
Changsha 410083, China
[2]The Hunan Provincial Key Lab of Bioinformatics, Central South University
Changsha 410083, China
[3]The College of Computer Science and Technology,China University of Petroleum
Qingdao 266580, China
[4]School of Information Science and Technology, University of Science and Technology of China
HeFei, 230026, China
[5]Institute of Artificial Intelligence, Hefei Comprehensive National Science Center
HeFei, 230026, China
`jxwang@mail.csu.edu.cn, jhxu@ustc.edu.cn`

## Abstract

Graph self-supervised learning (GSL) is essential for processing graph-structured data, reducing the need for manual labeling. Traditionally, this paradigm has extensively utilized Bayesian Personalized Ranking (BPR) as its primary loss function. Despite its widespread application, the theoretical analysis of its node relations evaluation have remained largely unexplored. This paper employs recent advancements in latent hyperbolic geometry to deepen our understanding of node relationships from a graph-theoretical perspective. We analyze BPR's limitations, particularly its reliance on local connectivity through 2-hop paths, which overlooks global connectivity and the broader topological structure. To address these shortcomings, we purpose a novel loss function, BPR+, designed to encompass even-hop paths and better capture global connectivity and topological nuances. This approach facilitates a more detailed measurement of user-item relationships and improves the granularity of relationship assessments. We validate BPR+ through extensive empirical testing across five real-world datasets and demonstrate its efficacy in refining graph self-supervised learning frameworks. Additionally, we explore the application of BPR+ in drug repositioning, highlighting its potential to support pharmaceutical research and development. Our findings not only illuminate the success factors of previous methodologies but also offer new theoretical insights into this learning paradigm.

## 1 Introduction

Self-supervised learning (SSL) is revolutionizing deep learning by significantly reducing the reliance on manually annotated labels [1]. This approach has gained considerable traction, especially in the application to graph-structured data, referred to as graph self-supervised learning (GSL) [2]. GSL exploits the unique, complex topological structures of nodes (non-Euclidean data space) to generate embeddings [2]. This contrasts sharply with the more straightforward Euclidean spaces typical of image and language data, presenting unique challenges and opportunities for innovation [1].

---

*Corresponding authors

In GSL, designing an effective loss function capable of navigating the irregularities of graph data is crucial. Bayesian Personalized Ranking (BPR) has been widely adopted in GSL methods to address this need [3]. Originally formulated to solve the personalized ranking problem in recommendation systems, BPR assesses item rankings for a user based on their historical interactions [4]. The BPR loss function measures the relationships between users and items by calculating the dot product of their embeddings, differentiating connected pairs (positive samples) from unconnected pairs (negative samples) in the graph [5].

Despite BPR loss widespread use, the theoretical analysis of its relations evaluation within GSL remains a relatively unexplored area. In this paper, we analyze the physical underpinnings behind BPR loss from a graph theoretical perspective. Notably, based on the formula for BPR loss, the score $y_{ui}$ can be interpreted as the computation of elements in the matrix $\boldsymbol{E}\boldsymbol{E}^\top$, specifically $y_{ui} = (\boldsymbol{E}\boldsymbol{E}^\top)_{ui}$. If we consider $\boldsymbol{E}$ as an embedding network comprising users, items, and abstract nodes, then $y_{ui}$ can be understood as a statistic of the 2-hop paths from user $u$ to item $i$ within this network. However, these 2-hop paths only capture local connectivity information. This coarse-grained assessment approach brings about two issues. Firstly, it lacks precision in measuring topological similarity, resulting in a single score corresponding to a wide range of topological similarities (e.g., from 20% to 50%). Secondly, these scores are influenced by the norm of embedding vectors, higher norm correspond to higher scores with other embeddings.

To address these limitations, it seems intuitive to extend our analysis to include longer n-hop paths. However, the selection of these paths and the quantification of their informational content pose formidable challenges. Previous study has shown that when the graph is sufficiently large, 2-hop paths approximate the concept of energy distance defined in latent hyperbolic geometry [6], derived using the principle of maximum entropy [7]. This measures the complement of the ratio of the latent distance between nodes in latent hyperbolic space to their chemical potential (a function of expected degree). Previous work indicates that as a measure of energy distance, 2-hop paths are not sufficiently precise. Consequently, we introduce the TopoLa distance to measure energy distance between nodes and, based on this, propose the BPR+ loss. We demonstrate that BPR+ loss more accurately measures the topological structure similarity and global connectivity information between users and items in the embedding network.

In addition to improve BPR loss, the introduction of network geometry (latent hyperbolic geometry) allows us to analyze common graph convolution operations from a new perspective and explain their effectiveness [6]. This not only clarifies the reasons behind the success of previous graph self-supervised learning but also provides a fresh perspective for the theoretical analysis of this learning paradigm. Extensive empirical tests of five existing methods on five real datasets confirm our theory and demonstrated the effectiveness of our new approach. Additionally, we apply BPR+ to the field of pharmaceutical research and development, developing a drug repositioning framework, TopoDR, and selecting ten potential therapeutic candidates for four prevalent cancers: colorectal, breast, stomach cancer, and leukemia.

**Contributions and paper structure.** We analyze Bayesian Personalized Ranking (BPR) loss from a graph-theoretical perspective, with a particular focus on the physical significance and its impact on embedding representations. In this process, we demonstrate that the measurement of relationships between users and items can be transformed into a path problem in graph theory, allowing us to use network geometry to aid the analysis. The main contributions are as follows:

•Demonstrate the limitations of BPR loss through latent hyperbolic geometry and provide relevant proofs (Section 3).

•Explore the effectiveness of common graph convolution operations in existing graph neural networks from the perspective of topology-encoded latent hyperbolic geometry topologically (Section 4).

•Propose a novel loss function, BPR+, as an alternative (Section 5).

•Develop an approximation technique based on numerical analysis to optimize the computation of loss. (Section 6).

•Apply BPR+ to the field of pharmaceutical research and development, introducing a novel computational framework for drug repositioning (Appendix F).

## 2 Preliminaries

**Bayesian Personalized Ranking.** BPR is a pairwise loss designed for one-class collaborative filtering [3]. The objective function of BPR encourages higher predictions for observed entries relative to their unobserved counterparts, formulated as follows:

$$L_{BPR} = -\sum_{u=1}^{M} \sum_{i \in \mathcal{N}_u} \sum_{j \notin \mathcal{N}_u} \ln \sigma \left( \hat{y}_{ui} - \hat{y}_{uj} \right) + \tau \left\| \mathbf{E}^{(0)} \right\|^2 \tag{1}$$

where $\mathcal{N}_u$ represents the set of items associated with user $u$. The parameter $\tau$ is utilized to regulate the intensity of L2 regularization. $M$ denotes the number of users. The variable $\hat{y}_{ui}$ is obtained through the product of the embeddings of user $u$ and item $i$, that is $\hat{y}_{ui} = \boldsymbol{e}_u \boldsymbol{e}_i^\top$.

## 3 The limitation of BPR loss

In the Bayesian Personalized Ranking (BPR) loss, the score $\hat{y}_{ui}$ between the user $u$ and the item $i$ is derived from the dot product of the embeddings of user $u$ and item $i$. We find that computing $\hat{y}_{ui}$ is equivalent to calculating the Gram matrix $\boldsymbol{E}\boldsymbol{E}^\top$ of the embedding matrix $\boldsymbol{E}$. In graph theory, the Gram matrix $\boldsymbol{E}\boldsymbol{E}^\top$ represents the number of 2-hop paths between user $u$ and item $i$. During $\hat{y}_{ui}$ computation, we model matrix $\boldsymbol{E}$ as a graph comprising users, items, and $N_e$ abstract nodes (Figure 1).

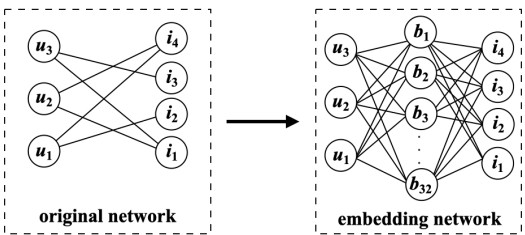

Figure 1: The illustration of embedding network.

Thus, graph self-supervised learning can be understood as the process of constructing an embedding network with a specified number of abstract nodes along with users and items to preserve the information of the original network. The weights of the edges in this network represent the relationships between the abstract nodes and other elements in the network. In BPR loss, $\hat{y}_{ui}$ evaluates the interaction between user $u$ and item $i$ by assessing 2-hop paths via abstract nodes (Figure 2). Krioukov has demonstrated through the principle of maximum entropy that the counting of 2-hop paths positively correlates with the complement of the energy distance between nodes in the latent hyperbolic [7]. However, Previous research indicates that relying solely on this local connectivity information is insufficient for accurately measuring the energy distance between nodes in a latent hyperbolic space, as stated in **Theorem 1** (Appendix A). According to Theorem 1, we demonstrate that there exists a more precise method for measuring energy distance, which surpasses the method of counting 2-hop paths.

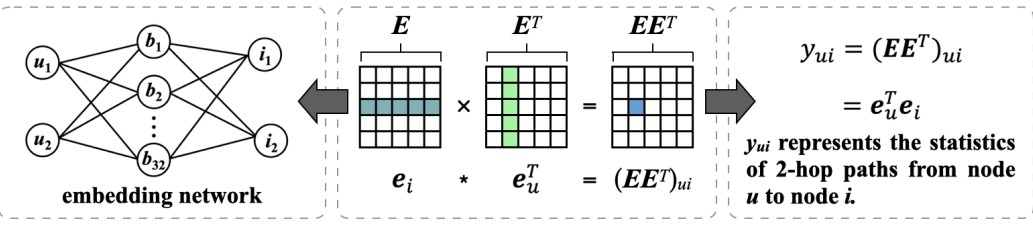

Figure 2: The physical significance of the Bayesian Personalized Ranking (BPR) loss.

**Theorem 1.** *Given a network $X \in \mathbb{R}^{(m+n) \times N_e}$, let $\langle t(x) \rangle$ denote the expected number of triangles, and $\langle t'(x) \rangle$ represent the expected number of weighted triangles. Consequently, $\alpha' > \alpha$, where $\alpha'$ is the logarithm of the thermodynamic activity corresponding to $\langle t'(x) \rangle$, and $\alpha$ is associated with $\langle t(x) \rangle$.*

## 4    Analyzing common graph convolution operations using network geometry

As shown in the previous section, the embedding matrix can be regarded as an embedding network. Consequently, we can utilize network geometry to analyze common graph convolution operations and explain their effectiveness. We use lightGCN as an example, a widely utilized foundational module.

**Light Graph Convolution (LGC).** Within the lightGCN, LGC is designed to focus on the fundamental elements of graph convolution, specifically information propagation and aggregation, while forgoing superfluous feature transformations and nonlinear activation functions. The operation streamlines the graph convolution process, enhancing the efficiency. The formulation of LGC is defined as follows:

$$e_u^{(k+1)} = \sum_{i \in \mathcal{N}_u} \frac{1}{\sqrt{|\mathcal{N}_u|}\sqrt{|\mathcal{N}_i|}} e_u^{(k)}, \qquad e_i^{(k+1)} = \sum_{u \in \mathcal{N}_i} \frac{1}{\sqrt{|\mathcal{N}_i|}\sqrt{|\mathcal{N}_u|}} e_i^{(k)}, \qquad (2)$$

where the symmetric normalization factor $\frac{1}{\sqrt{|\mathcal{N}_r|}\sqrt{|\mathcal{N}_d|}}$, in line with the conventional Graph Convolutional Network (GCN) architecture, is employed to curb the increase in embedding scale ensuing from successive graph convolution operations. The embedding of the $k$-th layer can be calculated as follows:

$$\boldsymbol{E}^{(k)} = \alpha_k \left( \boldsymbol{D}^{-\frac{1}{2}} \boldsymbol{A} \boldsymbol{D}^{-\frac{1}{2}} \right)^k \boldsymbol{E}^{(0)}, \qquad (3)$$

where $\alpha_k \geq 0$ denotes the importance of the embeddings from the $k$-th layer in constituting the final embedding. $\boldsymbol{D} \in \mathbb{R}^{(m+n) \times (m+n)}$ represents the degree matrix, where $\boldsymbol{D}_{ii}$ indicates the degree of the $i$-th node in the adjacency matrix $\boldsymbol{A}$. From a graph theory perspective, the $k$-th layer of graph convolution symbolizes the single-step diffusion of the ID embedding $\boldsymbol{E}^{(0)}$, which is influenced by $k$-hop paths connecting user $u$ and item $i$ within $\boldsymbol{A}$. The symmetric normalization factor weights the statistics of $k$-hop paths to address their exponential increase.

**Layer Combination.** In the lightGCN architecture, the sole trainable elements are the ID embeddings: $e_u^{(0)}$ and $e_i^{(0)}$. Post-initialization, embeddings obtained at $k$-th layer are calculated via LGC, as defined in Equation (3). Subsequently, the final network embedding are constructed through a weighted summation of embeddings from each layer:

$$\boldsymbol{e}_u = \sum_{k=0}^{K} \alpha_k \boldsymbol{e}_u^{(k)}, \qquad \boldsymbol{e}_i = \sum_{k=0}^{K} \alpha_k \boldsymbol{e}_i^{(k)}, \qquad (4)$$

where $K$ is the number of layers. The matrix form of layer combination can be defined as:

$$\begin{aligned} \boldsymbol{E} &= \alpha_0 \boldsymbol{E}^{(0)} + \alpha_1 \boldsymbol{E}^{(1)} + \alpha_2 \boldsymbol{E}^{(2)} + \cdots \alpha_K \boldsymbol{E}^{(K)} \\ &= \alpha_0 \boldsymbol{E}^{(0)} + \alpha_1 \widetilde{\boldsymbol{A}} \boldsymbol{E}^{(0)} + \alpha_2 \widetilde{\boldsymbol{A}}^2 \boldsymbol{E}^{(0)} + \cdots \alpha_K \widetilde{\boldsymbol{A}}^K \boldsymbol{E}^{(0)} \end{aligned} \qquad (5)$$

where $\widetilde{\boldsymbol{A}} = \boldsymbol{D}^{-\frac{1}{2}} \boldsymbol{A} \boldsymbol{D}^{-\frac{1}{2}}$ is the symmetrically normalized matrix. From a graph theory perspective, there are three reasons why layer combination is effective: (1) With an increase in hops, the variance in the number of paths between nodes diminishes, leading to a reduction in the disparity between layer embeddings and explaining the tendency for over-smoothing at higher layers. (2) As demonstrated in previous work, path counts encapsulate information about node degrees and topological similarities between nodes. However, the extent to which these factors are reflected varies with the hop count, and fusing embeddings from multiple layers effectively captures this nuanced information. (3) This

operation has the same effect as self-connection in graph convolution. Essentially, self-connection adds a path connecting each node to itself.

## 5 The Improved Bayesian Personalized Ranking (BPR+) loss.

In the graph self-supervised learning (GSL) framework, abstract nodes are introduced to reconstruct the original network, preserving existing node relationships. The Bayesian Personalized Ranking (BPR) loss evaluates user-item relationships in this restructured network based on 2-hop path statistic. Previous work has demonstrated that 2-hop paths serve as approximate representations of energy distance in the latent hyperbolic space [8]. Our research, based on the maximum entropy principle, demonstrates that weighting 2-hop paths aligns more closely with the actual energy distance, revealing BPR loss's limitations. This led to the introduction of TopoLa distance, defined as follows:

$$d_{\text{topo}}(u, i) = \frac{1}{\lambda}|2 - \text{hop}| - \frac{1}{\lambda^2}|4 - \text{hop}| + \frac{1}{\lambda^3}|6 - \text{hop}| - \cdots \tag{6}$$

where $\lambda$ is the scale factor. The BPR+ loss (Figure 3) can be represented as:

$$L_{BPR+} = -\sum_{u}^{m} \sum_{i \in \mathcal{N}_u} \sum_{j \notin \mathcal{N}_u} \ln \sigma \left( \frac{d_{\text{topo}}(u, i) - d_{\text{topo}}(u, j)}{\lambda} \right) + \tau \left\| \boldsymbol{E}^0 \right\|^2. \tag{7}$$

This incorporation of all even-hop paths offers two advantages: (1) global information is introduced to amalgamate relational information of all users and items in the network; (2) The TopoLa distance captures the topological structure information between users and items (**Theorem 2** and **Theorem 3**). See Appendix B and C for details.

**Theorem 2.** *Given a user embedding $y \in \mathbb{R}^{N_e}$, the embedding matrix $\boldsymbol{E} \in \mathbb{R}^{(m+n) \times N_e}$, and a parameter $\lambda$, the following optimal solution to the problem is denoted by $\boldsymbol{c}^*$ in vector form:*

$$\min \frac{1}{\lambda} \|\boldsymbol{y} - \boldsymbol{c}\boldsymbol{E}\|_F^2 + \|\boldsymbol{c}\|_F^2 \tag{8}$$

*We have*

$$\frac{\left\| \boldsymbol{c}_i^* - \boldsymbol{c}_j^* \right\|_F^2}{\|\boldsymbol{y}\|_F^2} \leq \frac{1}{\lambda} \left\| \boldsymbol{e}_i^\top - \boldsymbol{e}_j^\top \right\|_F^2 \tag{9}$$

**Theorem 3.** *Given nodes $i$ and $j$, the topological similarity between them is directly proportional to the $d_{topo}$ value.*

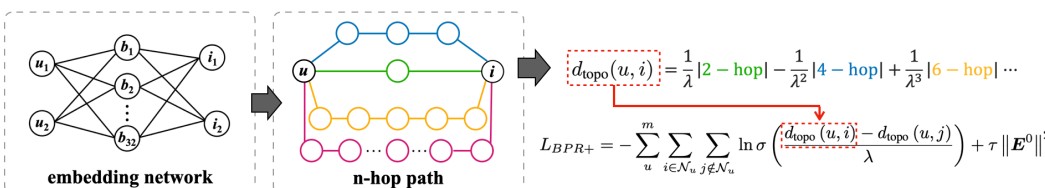

Figure 3: The illustration of BPR+ loss.

## 6 Optimizing loss computation

Direct computation of the loss using the TopoLa distance from the previous section, which involves multiple matrix multiplications, is computationally inefficient. To address this, we have optimized the computational approach to enhance efficiency. The matrix power form of the TopoLa distance can be defined as:

$$\boldsymbol{D}_{\text{topo}} = \frac{1}{\lambda}\boldsymbol{H} - \frac{1}{\lambda^2}\boldsymbol{H}^2 + \frac{1}{\lambda^3}\boldsymbol{H}^3 - \frac{1}{\lambda^4}\boldsymbol{H}^4 \cdots \tag{10}$$

where $\boldsymbol{H} = \boldsymbol{E}\boldsymbol{E}^\top$ is the 2-hop path statistic matrix of $\boldsymbol{E}$. $d_{\text{topo}}(u, i)$ is an element of $\boldsymbol{D}_{\text{topo}}$, denoted as $(\boldsymbol{D}_{\text{topo}})_{ui} = d_{\text{topo}}(u, i)$. Due to the necessity of multiple matrix multiplications, the time complexity for calculating matrix $\boldsymbol{D}_{\text{topo}}$ is $O\left(N_b^3\right) \cdot N_b$ is the batch size. To enhance computational efficiency, we employ matrix decomposition techniques. Firstly, the embedding matrix $\boldsymbol{E}$ undergoes singular value decomposition (SVD), represented as $\boldsymbol{E} = \boldsymbol{U}\boldsymbol{\Sigma}\boldsymbol{V}^\top$. Consequently, the formula for $\boldsymbol{D}_{\text{topo}}$ can be expressed as:

$$\boldsymbol{D}_{\text{topo}} = \boldsymbol{U}\left(\frac{1}{\lambda}\boldsymbol{\Sigma}^2 - \frac{1}{\lambda^2}\boldsymbol{\Sigma}^4 + \frac{1}{\lambda^3}\boldsymbol{\Sigma}^6 - \frac{1}{\lambda^4}\boldsymbol{\Sigma}^8 \cdots\right)\boldsymbol{U}^\top. \tag{11}$$

As a result, the time complexity is reduced to $\mathcal{O}(N_b N_e^2 + N_b^2 N_e + N_e^3)$. $N_e$ is the embedding size. Given that in GSL models, $N_b$ significantly exceeds $N_e$, the proposed calculation form offers a lower time complexity.

## 7    Experiments

To confirm the superiority and efficacy of the proposed BPR+ loss, we carried out extensive experiments to explore the following research questions:

•**RQ1**: How much improvement can our proposed loss bring when applied to various state-of-the-art GSL models?

•**RQ2**: How do the physical properties of our proposed loss differ from those of Bayesian Personalized Ranking (BPR) loss?

•**RQ3**: How can numerical analysis enhance the efficiency of loss computation?

•**RQ4**: How does our proposed technique perform in practical applications?

### 7.1    Experimental Settings

**Evaluation Datasets**. In this section, we evaluate our model and several baselines on five real-world datasets: Amazon, Gowalla, Yelp, LastFM, and Beer. The Amazon dataset includes implicit feedback from users on books from the Amazon platform. The Gowalla dataset, derived from the Gowalla platform, records user check-ins at different locations, with data provided for January to June 2010. The Yelp dataset contains user ratings and interaction data for various businesses. The LastFM dataset comprises social network interactions, tags, and music artist listening information from the Last.fm online music system. The Beer dataset, sourced from the BeerAdvocate platform, includes beer reviews and is filtered to only retain users and items with at least ten interactions. Details and statistics are provided in Table 1.

Table 1: Statistics of the experimental datasets.

| Dataset | User # | Item # | Interaction # | Density |
|---------|--------|--------|---------------|---------|
| Amazon | 76,469 | 83,761 | 966,680 | $1.5 \times 10^{-4}$ |
| Gowalla | 25,557 | 19,747 | 294,983 | $5.8 \times 10^{-4}$ |
| Yelp | 42,712 | 26,822 | 182,357 | $1.6 \times 10^{-4}$ |
| LastFM | 1,892 | 17,632 | 92,834 | $2.8 \times 10^{-3}$ |
| Beer | 10,456 | 13,845 | 1,381,094 | $9.5 \times 10^{-3}$ |

**Evaluation Protocols**. We downloaded the datasets from prior work, including its predefined training, validation, and test sets. The evaluation metrics used include Recall@N and Normalized Discounted Cumulative Gain (NDCG)@N , where N = {10, 20}, both of which are better when higher.

**Baseline Methods**. We evaluated the general applicability of the proposed technique by applying BPR+ loss to various baseline models. The details of these baselines are provided below.

•LightGCN [9]: It simplifies the traditional Graph Convolution Network (GCN) for collaborative filtering by focusing solely on neighborhood aggregation, eschewing feature transformations and nonlinear activations to enhance training efficiency and improve recommendation performance.

•SGL [10]: It improves LightGCN by using contrastive learning with data augmentation techniques like random walks and node/edge dropout, enhancing accuracy and robustness.

•NCL [11]: It is a neighborhood-enriched contrastive learning approach that boosts the effectiveness of graph collaborative filtering by incorporating structural and semantic neighbors into contrastive pairs, enhancing the model's ability to handle data sparsity and improve recommendation accuracy.

•LightGCL [12]: It is a graph contrastive learning approach that enhances graph-based recommender systems by using singular value decomposition for data augmentation, effectively preserving the intrinsic semantic structures and offering robustness against noise perturbation and data sparsity.

•AdaGCL [5]: It is a graph contrastive learning framework that improves collaborative filtering through two trainable view generators: a graph generative model and a graph denoising model, which create adaptive contrastive views to effectively address data sparsity and noise issues.

## 7.2 Overall Performance Validation (RQ1)

The effectiveness of the proposed BPR+ loss is validated by applying it to various baseline models and conducting a comprehensive performance evaluation across five datasets. We present the experimental result in Table 2. The findings indicate a consistent pattern: algorithms incorporating BPR+ loss exhibit improved performance across all datasets and metrics. For instance, in the Amazon dataset, compared to adaGCL, adaGCL+ achieves an 11.9% improvement in the Recall@10 metric; for the NDCG@10 metric, the increase reaches 13.4%. The consistent improvement across almost all datasets and algorithms highlights the versatility and robustness of the BPR+ loss. This enhancement not only leads to higher recall rates, meaning more relevant items are recommended, but also improves the NDCG scores, indicating a better ranking quality in the recommendations provided. See Appendix D for details. In addition, see Appendix E for hyperparameter analysis.

Table 2: Performance comparison on Amazon, Gowalla, Yelp, LastFM, Beer datasets in terms of Recall and NDCG.

| Dataset | Metric | adaGCL | adaGCL+ | lightGCL | lightGCL+ | NCL | NCL+ | SGL | SGL+ | lightGCN | lightGCN+ |
|---|---|---|---|---|---|---|---|---|---|---|---|
| Amazon | Recall@10 | 0.0612 | **0.0685** | 0.0576 | **0.0608** | 0.0736 | **0.0783** | 0.0432 | **0.0454** | 0.0535 | **0.0722** |
| | NDCG@10 | 0.0507 | **0.0575** | 0.0476 | **0.0510** | 0.0608 | **0.0656** | 0.0269 | **0.0282** | 0.0439 | **0.0607** |
| | Recall@20 | 0.0946 | **0.1032** | 0.0892 | **0.0942** | 0.1125 | **0.1166** | 0.0678 | **0.0690** | 0.0838 | **0.1080** |
| | NDCG@20 | 0.0620 | **0.0691** | 0.0583 | **0.0621** | 0.0739 | **0.0784** | 0.0343 | **0.0351** | 0.0541 | **0.0727** |
| Gowalla | Recall@10 | 0.1523 | **0.1536** | 0.1690 | **0.1693** | 0.1755 | **0.1768** | 0.1641 | **0.1650** | 0.1544 | **0.1564** |
| | NDCG@10 | 0.1206 | **0.1208** | 0.1359 | **0.1365** | 0.1409 | **0.1420** | 0.1321 | **0.1323** | 0.1248 | **0.1264** |
| | Recall@20 | 0.2253 | **0.2264** | 0.2444 | **0.2451** | 0.2557 | **0.2571** | 0.2369 | 0.2369 | 0.2297 | **0.2317** |
| | NDCG@20 | 0.1427 | **0.1432** | 0.1585 | **0.1592** | 0.1649 | **0.1663** | 0.1540 | **0.1541** | 0.1474 | **0.1489** |
| Yelp | Recall@10 | 0.0517 | **0.0529** | 0.0456 | **0.0520** | 0.0503 | **0.0510** | 0.0654 | **0.0739** | 0.0505 | **0.0525** |
| | NDCG@10 | 0.0320 | **0.0334** | 0.0284 | **0.0330** | 0.0313 | **0.0318** | 0.0554 | **0.0742** | 0.0314 | **0.0325** |
| | Recall@20 | 0.0805 | **0.0837** | 0.0716 | **0.0830** | **0.0794** | 0.0790 | 0.0978 | **0.1055** | **0.0798** | 0.0796 |
| | NDCG@20 | 0.0406 | **0.0425** | 0.0362 | **0.0421** | 0.0398 | **0.0400** | 0.0662 | **0.0863** | 0.0400 | **0.0405** |
| LastFM | Recall@10 | 0.1688 | **0.1760** | 0.1403 | **0.1412** | 0.1772 | **0.1775** | 0.1655 | **0.1734** | 0.1511 | **0.1715** |
| | NDCG@10 | 0.1521 | **0.1553** | **0.1243** | 0.1239 | 0.1562 | **0.1566** | 0.1470 | **0.1526** | 0.1343 | **0.1529** |
| | Recall@20 | 0.2465 | **0.2499** | 0.1949 | **0.1964** | 0.2525 | **0.2538** | 0.2433 | **0.2486** | 0.2206 | **0.2480** |
| | NDCG@20 | 0.1843 | **0.1860** | **0.1475** | 0.1472 | 0.1877 | **0.1883** | 0.1790 | **0.1836** | 0.1634 | **0.1847** |
| Beer | Recall@10 | 0.0448 | **0.0457** | 0.0320 | **0.0324** | 0.0729 | **0.0734** | 0.0730 | **0.0731** | 0.0670 | **0.0757** |
| | NDCG@10 | 0.0584 | **0.0608** | 0.0533 | **0.0538** | 0.0922 | **0.0923** | 0.0922 | 0.0922 | 0.0857 | **0.0937** |
| | Recall@20 | 0.0731 | **0.0757** | 0.0489 | **0.0491** | 0.1171 | **0.1175** | 0.1165 | **0.1167** | 0.1084 | **0.1227** |
| | NDCG@20 | 0.0647 | **0.0675** | 0.0542 | **0.0547** | 0.1013 | **0.1014** | 0.1009 | 0.1009 | 0.0938 | **0.1039** |

## 7.3 Physical Property Analysis (RQ2)

In this section, we explore the physical properties of BPR loss and BPR+ loss. To simulate ID embeddings, a 1000×64 matrix is constructed with its elements set to 0 and 1. Our first evaluation focused on whether these loss functions could accurately reflect the topological similarity between users and items, as calculated using the Jaccard index. Observations from Figures 4A and 4B demonstrate that the relationships derived by both loss functions proportionally align with topological similarities. However, BPR exhibits a coarse granularity, where different topological similarities may correspond to identical scores.

Further analysis involved examining the relationship between the number of neighbors in a user-item pair and the relationship scores. The number of neighbors indicates the norm of the user/item embeddings. The union of neighbors represents the number of non-zero elements in the sum of two

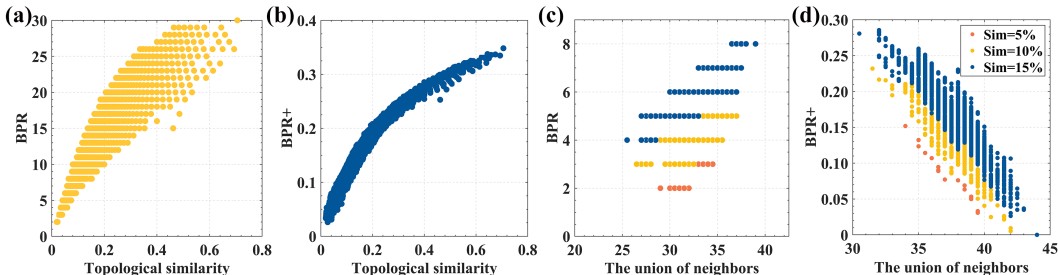

Figure 4: Comparison of the physical properties of BPR loss and BPR+ loss.

embedding vectors. A user connected to numerous abstract nodes will have a larger norm, leading to higher dot products with all item embeddings. Our experimental validation supports this analysis: The BPR loss scores relationships, increasing with the expanding union of neighbors. Additionally, when classifying these scores by varying degrees of topological similarity (within a range of ±2%), we observe that BPR delineates less distinct boundaries between different similarity levels with a narrower range of scores. In contrast, BPR+ offers a more nuanced evaluation, assigning distinct scores to different degrees of similarity, thus enhancing the fidelity of relationship assessments.

## 7.4 Efficiency Study (RQ3)

Due to the inefficiency of calculating loss directly via matrix exponentiation, we incorporate matrix factorization techniques to accelerate the loss computation process. To analyze computational efficiency in practical scenarios, we select the NCL as an example and compared the computation times of three loss computation methods: BPR, BPR+ (MM), and BPR+ (MF), across five datasets, as illustrated in Figure 5A. We record the time required to complete one epoch for each loss function. BPR+ (MM) denotes BPR+ calculated using matrix multiplication, whereas BPR+ (MF) refers to BPR+ calculated using matrix factorization, both considering 40-hop paths. The results indicate that using the optimized loss computation methods significantly reduces computation time, comparable to that of BPR. For instance, on the Amazon dataset, the computation time for BPR is 30.5s, for BPR+(MF) it is 36.5s, and for BPR+(MM), it significantly increases to 227.7s. Moreover, the increase in data scale further enhances the improvement in computational efficiency. Similarly, the results for LightGCL on the same datasets are consistent (Figure 5B).

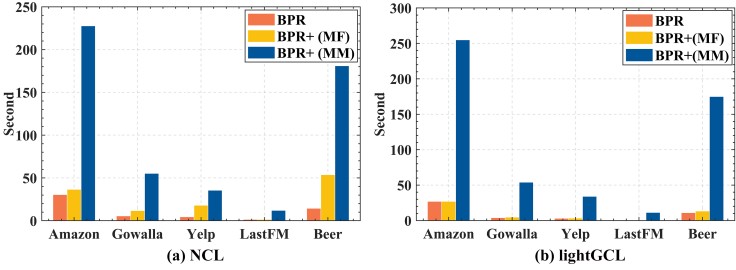

Figure 5: Comparison of computational efficiency between BPR, BPR+(MM), and BPR+(MF).

## 7.5 Case Study (RQ4)

In this section, we apply our proposed technique specifically to the domain of drug repositioning to validate its performance in practical scenarios. We develop a new framework to aid pharmaceutical research and development (TopoDR), integrating multimodal drug and disease information encompassing chemical structures, side effects, drug-drug interactions, drug target profiles, disease phenotypes, and disease ontologies.

To verify the validity of the framework, we compare TopoDR with four leading drug repositioning methods (deepDR [13], DRHGCN [14], DDAGDL [15], and AdaDR [16]) using their default settings

from public codes, validated on gold standard datasets (Fdataset and Cdataset). Performance was assessed via 10-fold cross-validation, with results detailed in Figure 6. We use six evaluation metrics to compare the performance of TopoDR with other methods: the area under the receiver operating characteristic (ROC) curve (AUC), the area under the precision-recall curve (AUPR), accuracy (Acc.), precision (Pre.), recall (Rec.), and Matthew's correlation coefficient (MCC). In these comparisons, particularly on the Fdataset, TopoDR consistently outperformed the other models in metrics assessing overall model performance. Specifically, in the Fdataset, TopoDR's AUC, AUPR, and MCC are 0.9594, 0.9630, and 0.7440, respectively, surpassing AdaDR, the next best method, by 2.89%, 1.86%, and 7.34%. Similarly, on the Cdataset, TopoDR exceeds AdaDR by 1.82%, 1.18%, and 2.64%. These results confirm TopoDR's superior ability to accurately predict potential indications for drugs. Furthermore, we select ten potential therapeutic candidates for four prevalent cancers (colorectal, breast, stomach cancer, and leukemia). Table 6 displays the top ten candidate drugs for these cancers, with confirmed therapeutic drugs from the Comparative Toxicogenomics Database (CTD) emphasized in bold. See Appendix F for details.

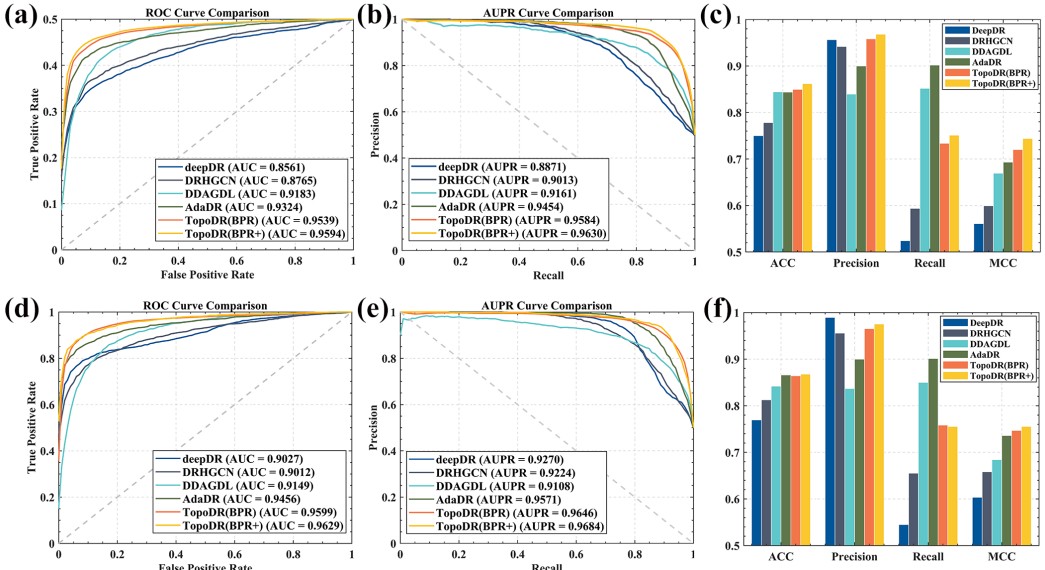

Figure 6: The performance of drug repositioning methods on Fdataset and Cdataset. Fdataset: (A) The ROC curve. (B) The precision-recall curve. (C) The accuracy, precision, recall, and Matthew's correlation coefficient of all methods. Cdataset: (D) The ROC curve. (E) The precision-recall curve. (F) The accuracy, precision, recall, and Matthew's correlation coefficient of all methods.

## 8 Conclusion

Although Bayesian Personalized Ranking (BPR) loss has been widely applied in graph self-supervised learning with considerable success, it currently lacks robust theoretical support and analysis for node relations evaluation. In this paper, we explain from a graph-theoretical perspective that BPR essentially counts the 2-hop paths from users to items within the embedding network. Analysis based on the principle of maximum entropy reveals that while BPR loss can represent the energy distance between users and items, it is not sufficiently precise. Based on these limitations, we propose a new loss function, BPR+, as an alternative. Our work not only elucidates the success of previous graph self-supervised learning efforts but also provides a fresh perspective for theoretical analysis in this field.

This work holds broad prospects. Foremost and most challenging is the inspiration derived from our theoretical analysis to design superior graph neural network modules. Additionally, our work represents a step towards a deeper understanding of the relationships between nodes, which is central to the utility of graph neural networks. However, computational time remains an unresolved issue. Therefore, accelerating computation will be a crucial direction for our future work.

# 9 Acknowledgments

This work was supported in part by the National Natural Science Foundation of China (62350004, 62332020, U22A2041, 62502540, 6250072467), Shandong Provincial Natural Science Foundation (ZR2025QC1546, ZR2025QC1540), Fundamental Research Funds for the Central Universities (25CX06033A), the Project of Xiangjiang Laboratory (No. 23XJ01011), the Science and Technology Major Project of Changsha (No.kh2402004), Hunan Provincial Postgraduate Scientific Research Innovation Project (CX20230234), the start-up funds from USTC and IAI.

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

# A  Proof of Theorem 1

We restate Theorem 1: *Given a network $X \in \mathbb{R}^{(m+n) \times N_e}$, let $\langle t(x) \rangle$ denote the expected number of triangles, and $\langle t'(x) \rangle$ represent the expected number of weighted triangles. Consequently, $\alpha' > \alpha$, where $\alpha'$ is the logarithm of the thermodynamic activity corresponding to $\langle t'(x) \rangle$, and $\alpha$ is associated with $\langle t(x) \rangle$.*

***Proof.*** In network geometry, the expected numbers of triangles can be obtained through the integration of the graphon [7] :

$$\langle t(x) \rangle = \frac{1}{2} \iint_{\mathbb{R}^2} p(x,y)p(y,z)p(z,x)dydz, \tag{12}$$

Under the assumption of redundancy in the current statistics, the non-redundant expectation $\langle t'(x) \rangle$, is anticipated to be less than its redundant counterpart, $\langle t(x) \rangle$. In this context, $t(x)$ denotes the quantity of triangles associated with node $x$. Utilizing the maximum-entropy principle (21), we derive the following formula:

$$\langle t'(x) \rangle = \frac{1}{2} \iint_{\mathbb{R}^2} p'(x,y)p'(y,z)p'(z,x)dydz = \bar{t}' < \bar{t}. \tag{13}$$

Therefore,

$$\frac{1}{2} \iint_{\mathbb{R}^2} p'(x,y)p'(y,z)p'(z,x)dydz < \frac{1}{2} \iint_{\mathbb{R}^2} p(x,y)p(y,z)p(z,x)dydz, \tag{14}$$

When the network size is sufficiently large, the approximate solution for the graphon that maximizes entropy is the Fermi-Dirac graphon:

$$p^*(x,y) = \begin{cases} \frac{1}{1+e^{2\alpha\left(r-\frac{1}{2}\right)}} & \text{if } 0 \leq r \leq 1, \\ \frac{1}{1+e^{\alpha}} & \text{if } r > 1, \end{cases} \tag{15}$$

where $\alpha$ and $r$ are the rescaled inverse temperature and energy distance, respectively. Inserting the terms from Formula [17] into Formula [16], we obtain:

$$\iint_{\mathbb{R}^2} \left[ \frac{1}{1+e^{2\alpha'\left(r-\frac{1}{2}\right)}} \right]^3 dydz < \iint_{\mathbb{R}^2} \left[ \frac{1}{1+e^{2\alpha\left(r-\frac{1}{2}\right)}} \right]^3 dydz. \tag{16}$$

Consequently, we deduce that $\alpha' > \alpha$. Based on prior research, the common neighbor integral corresponding to $\alpha'$ provides a more accurate representation of the latent space distance than that associated with $\alpha$ [7]. From this, we demonstrate that the current triangle statistics exhibit redundancy. Given that the common-neighbor integral is defined as $\int_{\mathbb{R}} \left[ \frac{1}{1+e^{2\alpha\left(r-\frac{1}{2}\right)}} \right]^2 dz$, the 2-hop paths (common neighbors) are also found to be imprecise when representing distances within the latent space and **Theorem 1** holds.

# B  Proof of Theorem 2

We restate Theorem 2: *Given a user embedding $y \in \mathbb{R}^{N_e}$, the embedding matrix $\boldsymbol{E} \in \mathbb{R}^{(m+n) \times N_e}$, and a parameter $\lambda$, the following optimal solution to the problem is denoted by $\boldsymbol{c}^*$ in vector form:*

$$\min \frac{1}{\lambda} \|\boldsymbol{y} - \boldsymbol{cE}\|_F^2 + \|\boldsymbol{c}\|_F^2 \tag{17}$$

*We have*

$$\frac{\left\| \boldsymbol{c}_i^* - \boldsymbol{c}_j^* \right\|_F^2}{\|\boldsymbol{y}\|_F^2} \leq \frac{1}{\lambda} \left\| \boldsymbol{e}_i^\top - \boldsymbol{e}_j^\top \right\|_F^2 \tag{18}$$

**Proof.** Inspired by prior work, we conducted the following inference [17]. Let $L(c) = \frac{1}{\lambda}\|\boldsymbol{y} - \boldsymbol{c}\boldsymbol{E}\|_F^2 + \|\boldsymbol{c}\|_F^2$. Since $\boldsymbol{c}^*$ is the optimal solution to Equation (17), it satisfies

$$\left.\frac{\partial L(c)}{\partial c_k}\right|_{c=c^*} = 0 \tag{19}$$

Thus, we have

$$-\frac{2}{\lambda}\boldsymbol{e}_i^\top\left(\boldsymbol{y} - \boldsymbol{c}^*\boldsymbol{E}\right) + 2\boldsymbol{c}_i^* = 0, \tag{20}$$

$$-\frac{2}{\lambda}\boldsymbol{e}_j^\top\left(\boldsymbol{y} - \boldsymbol{c}^*\boldsymbol{E}\right) + 2\boldsymbol{c}_j^* = 0. \tag{21}$$

Equation (20) and (21) give us

$$\boldsymbol{c}_i^* - \boldsymbol{c}_j^* = \frac{1}{\lambda}\left(\boldsymbol{e}_i^\top - \boldsymbol{e}_j^\top\right)\left(\boldsymbol{y} - \boldsymbol{c}^*\boldsymbol{E}\right). \tag{22}$$

Since $\boldsymbol{c}^*$ is optimal to equation (17), we get

$$\frac{1}{\lambda}\|\boldsymbol{y} - \boldsymbol{c}^*\boldsymbol{E}\|_F^2 + \|\boldsymbol{c}^*\|_F^2 = L\left(\boldsymbol{c}^*\right) \leq L(0) = \|\boldsymbol{y}\|_F^2 \tag{23}$$

Thus, we have $\frac{1}{\lambda}\|\boldsymbol{y} - \boldsymbol{c}^*\boldsymbol{E}\|_F^2 < \|\boldsymbol{y}\|_F^2$. Then equation (23) implies

$$\frac{\left\|\boldsymbol{c}_i^* - \boldsymbol{c}_j^*\right\|_F^2}{\|\boldsymbol{y}\|_F^2} \leq \frac{1}{\lambda}\left\|\boldsymbol{e}_i^\top - \boldsymbol{e}_j^\top\right\|_F^2 \tag{24}$$

The optimal solution $\boldsymbol{c}_i$ of optimization problem (17) corresponds to the $i$-th row of matrix $\boldsymbol{D}_{\text{topo}}$. From the above equation, we discern that the difference in distances between two nodes in the latent space relative to other nodes correlates with the divergence in their topological structures. That is, nodes with highly similar topological structures occupy proximate positions within the latent space and **Theorem 2** holds.

## C  Proof of Theorem 3

We restate Theorem 3:  *Given nodes $i$ and $j$, the topological similarity between them is directly proportional to the $d_{topo}$ value.*

**Proof.** Our way to exploring the relationship between Topola distance and the topological similarity is to differentiate various types of paths connecting nodes $i$ and $j$. By counting the number of each type of paths, it allows us to take into consideration not only local and global connectivity, but also node degrees in an implicit manner. Particularly, we focus on the classification of $n$-hop paths connecting nodes $i$ and $j$. Due to the possible existence of loop or loops on these paths, we establish a classification system for all such $n$-hop paths, based on the length of a path actually traversed (i.e., the number of hops after removing all the loops on the path). Clearly, there are a total of $(n-1)$ types. To facilitate a clear differentiation, we represent these path types with polygons. For example, a path that actually traverses 2-hop is defined as a $\mathrm{P}_2$ path, which can be viewed as a triangle if adding a direct edge back from node $j$ to node $i$. Similarly, a path that actually traverses 3 hops can be defined as $\mathrm{P}_3$ (represented as a quadrangle), and a path that actually traverses 4 hops can be defined as $\mathrm{P}_4$ (represented as a pentagon). The quantity of these types can be expressed as $|\mathrm{P}_l| = \text{loop}\left(a_l\right)$, where $a_l$ is the set of loop-free $l$-hop paths connecting nodes $i$ and $j$ and loop $(\cdot)$ represents the number of different ways of adding loops to paths in $a_l$ to form $n$-hop paths between nodes $i$ and $j$. While there is a correlation between $|\mathrm{P}_l|$ and $|a_l|$, accurately measuring this relationship remains challenging. Nevertheless, our observations suggest that a proportional relationship exists between the quantity of $\mathrm{P}_l$, characterized by solely loops between node $i$ and its neighbors, or between node $j$ and its neighbors, and $|a_l|$. Paths exhibiting such characteristics are defined as $b_n(l)$. The interplay between $|a_l|$ and $|b_n(l)|$ can be calculated as follows:

$$|b_n(l)| = \sum_{h=0}^{\frac{n-l}{2}} \kappa_i^h \kappa_j^{\frac{n-l}{2}-h} |a_l| \tag{25}$$

where $|b_n(l)|$ is the quantity of $b_n(l)$ in $n$-hop $P_l$ paths. $\kappa_i$ and $\kappa_j$ represent the degrees of node $i$ and node $j$. Thus, $n$-hop paths between node $i$ and node $j$ in the network are represented as:

$$
\begin{aligned}
\mid n - \text{ hop } \mid &= |a_n| + |b_n| + |c_n| \\
&= |a_n| + \sum_{P_l \in n- \text{ hop}} |b_n(P_l)| + |c_n| \\
&= \begin{cases} \sum_{t=1}^{\frac{n}{2}} \sum_{h=0}^{\frac{n}{2}-t} \kappa_i^h \kappa_j^{\frac{n}{2}-t-h} |a_{2t}| + |c_n| & \text{if } n \text{ is even} \\ \sum_{t=1}^{\frac{n-1}{2}} \sum_{h=0}^{\frac{n-1}{2}-t} \kappa_i^h \kappa_j^{\frac{n}{2}-t-h} |a_{2t+1}| + |c_n| & \text{if } n \text{ is odd} \end{cases}
\end{aligned}
\tag{26}
$$

where $c_n$ denotes the paths distinct from $a_n$ and $b_n$. $\mid n- \text{ hop}\mid$ refers to the counts of $n$-hop paths. Our analysis for 2-hop, 4-hop, and 6-hop paths reveals that $|c_n|$ does not exhibit a direct correlation with $\kappa_i$ and $\kappa_j$. Equation (26) reveals that $a_n$ and $b_n$ for even-hop paths consist of polygons with an odd number of sides, while for odd-hop paths, they comprise polygons with an even number of sides. Subsequently, we delve further into the physical significance of $\boldsymbol{D}_{\text{topo}}$, which can also be expressed in the form of a set of paths (note that the following formulas reveal the physical meanings of $\boldsymbol{D}_{\text{topo}}$ and $d_{\text{topo}}(i,j)$, but will not be used for calculation):

$$\boldsymbol{D}_{\text{topo}} = \frac{1}{\lambda} \boldsymbol{A}\boldsymbol{A}^\top - \frac{1}{\lambda^2} \boldsymbol{A}\boldsymbol{A}^\top\boldsymbol{A}\boldsymbol{A}^\top + \frac{1}{\lambda^3} \boldsymbol{A}\boldsymbol{A}^\top\boldsymbol{A}\boldsymbol{A}^\top\boldsymbol{A}\boldsymbol{A}^\top - \cdots \tag{27}$$

The emphasis on even-hop paths arises from their ability to capture the degree information and the global connectivity information of nodes $i$ and $j$. Furthermore, the TopoLa distance $d_{\text{topo}}$ is demonstrated to correlate with the topological structure of nodes. Specifically, $a_l$ can be classified into two types: those overlapping with 2-hop paths $(g_l)$ and the remaining ones $(s_l)$, with $|a_l| = |g_l| + |s_l|$. The $|s_l|$ relates to the topological structure of nodes $i$ and $j$. For instance, identical topological structure between $i$ and $j$ renders $|s_l|$ to zero (which indicates that it measures the topological similarity of nodes $i$ and $j$). Thus, we can express $d_{\text{topo}}$ as:

$$
\begin{aligned}
d_{\text{topo}}(i,j) &= \lim_{n\to\infty} \left[ \left( \sum_{t=1}^{\frac{n}{2}} \sum_{h=0}^{\frac{n}{2}-t} \frac{\kappa_i^h \kappa_j^{\left(\frac{n}{2}-t-h\right)}}{-(-\lambda)^{\left(\frac{n}{2}-t+1\right)}} \right) \left( \sum_{t=1}^{\frac{n}{2}} \frac{|a_{2t}|}{(-\lambda)^{t-1}} \right) + \sum_{t=1}^{\frac{n}{2}} \frac{c_{(2t)}}{\lambda^t} \right] \\
&= \lim_{n\to\infty} \left[ \left( \sum_{t=1}^{\frac{n}{2}} \sum_{h=0}^{\frac{n}{2}-t} \frac{\kappa_i^h \kappa_j^{\left(\frac{n}{2}-t-h\right)}}{-(-\lambda)^{\left(\frac{n}{2}-t+1\right)}} \right) \left[ \sum_{t=1}^{\frac{n}{2}} \frac{|g_{2t}|}{(-\lambda)^{t-1}} - \sum_{t=2}^{\frac{n}{2}} \frac{|s_{2t}|}{(-\lambda)^{t-1}} \right] + \sum_{t=1}^{\frac{n}{2}} \frac{c_{(2t)}}{\lambda^t} \right]
\end{aligned}
\tag{28}
$$

where $\sum_{t=1}^{\frac{n}{2}} \frac{|g_{2t}|}{(-\lambda)^{t-1}}$ and $\sum_{t=1}^{\frac{n}{2}} \frac{c_{(2t)}}{\lambda^t}$ are unaffected by node's topological structure, while $\sum_{t=2}^{\frac{n}{2}} \frac{|s_{2t}|}{(-\lambda)^{t-1}}$ acts as a topology-dependent penalty term. We find that the topological similarity between $i$ and $j$ is directly proportional to the $d_{\text{topo}}$ value and **Theorem 3** holds.

## D  More details about overall performance validation

The code for all models originates from SSLRec, with each model's embedding size fixed at 32 and the batch size set to 4096. To ensure a fair comparison, we employed a grid search method to determine the optimal parameter combination for each model, with $\lambda$ search range set to {1e-3, 1e-4, 1e-5, 1e-6, 1e-7}.

The Amazon, Gowalla, and Yelp datasets were downloaded from SSLRec [18] and have been pre-divided into training, validation, and test sets . The LastFM and Beer datasets were obtained from the corresponding code and data provided by adaGCL [5], and have also been pre-partitioned.

Experiments were conducted on a high-performance hardware platform comprising an Intel Xeon Platinum 8352V processor, an NVIDIA RTX 4090 with 24 GB of memory, and a system running

Ubuntu 20.04. The software environment included PyTorch version 1.11.0, Python 3.8, and CUDA 11.3.

To ensure statistical significance, we trained models using BPR loss and BPR+ loss, calculating the p-values. The results are presented in Table 3.

Table 3: Statistical significance of overall performance comparison.

| Dataset | adaGCL | lightGCL | NCL | SGL | lightGCN |
|---------|--------|----------|-----|-----|----------|
| Amazon | 1.97E-12 | 1.20E-18 | 1.34E-08 | 1.31E-13 | 7.37E-12 |
| Gowalla | 6.13E-11 | 3.29E-16 | 1.76E-13 | 3.72E-19 | 3.80E-14 |
| Yelp | 1.99E-10 | 5.31E-13 | 4.90E-11 | 2.78E-10 | 8.82E-08 |
| LastFM | 4.24E-12 | 2.38E-30 | 9.18E-10 | 1.65E-14 | 1.74E-11 |
| Beer | 7.56E-11 | 3.91E-10 | 8.44E-14 | 2.19E-15 | 3.17E-12 |

# E  Hyperparameter analysis

In this section, we explore the sensitivity of the adaGCL model to the critical hyperparameter $\lambda$ in the BPR+ loss formulation, which governs the influence of n-hop paths on relationship assessments. To ascertain the optimal $\lambda$, we systematically search across a range {1e-3, 1e-4, 1e-5, 1e-6, 1e-7}, aiming to determine how it impacts adaGCL's performance. As illustrated in Figure 7, the model's performance across five datasets with varying $\lambda$ values is analyzed. The data suggests that the optimal performance can occur at any point within the tested $\lambda$ range, highlighting the varying influence of n-hop paths on different datasets. This variability underscores the need for dataset-specific tuning of $\lambda$ to achieve the best results in adaGCL.

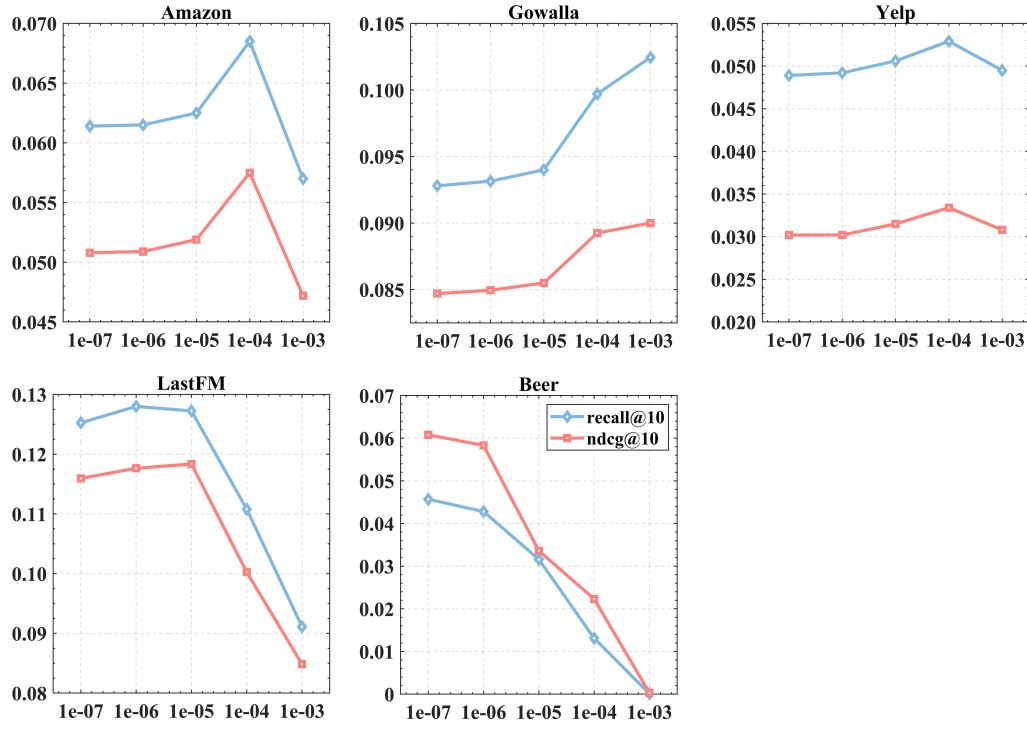

Figure 7: The impact of $\lambda$

# F  More details about case study

Drug repositioning, encompassing strategies like repositioning and therapeutic switching, is an evolving paradigm in pharmaceutical research, recognized for its time and cost efficiency compared

to traditional drug development [19, 20]. It leverages existing drugs, bypassing initial development phases, and accelerating market entry. Furthermore, drug repositioning emerges as a pragmatic way for addressing complex diseases like cancer [21, 22]. Computational drug repositioning utilizes molecular, clinical, and biophysical data to ascertain the potential of approved drugs for new clinical applications [23]. Such a strategy not only expedites the repositioning process but also minimizes associated costs [24]. Researchers have actively applied this strategy in identifying and developing therapeutic agents against COVID-19 [25].

With the advancement of artificial intelligence technology and the accumulation of large-scale biomedical data, drug repositioning based on deep learning (DL) has shown significant advantages over traditional computational approaches. For instance, DeepDR employed a collective variational autoencoder to integrate diverse network data, enhancing drug repositioning by effectively capturing non-linear structures and mitigating data sparsity [13]. Another innovative technique, DRHGCN, integrated data from various networks to refine drug repositioning [14]. It extracted both intra-domain and inter-domain features, improving drug and disease representations through sophisticated layer attention mechanisms, thereby enhancing repositioning accuracy. Additionally, DDAGDL, a framework that incorporated diverse drug-related networks, employed non-Euclidean domain knowledge and an attention mechanism to strengthen feature representation [15]. This framework further used XGBoost for robust prediction tasks. Lastly, AdaDR employed graph neural networks and attention mechanisms to integrate node features with topological structures [16]. Self-Supervised Learning (SSL) represents a potent paradigm that diminishes the dependency on manually annotated labels, facilitating training on extensive unlabeled datasets via self-supervised tasks [26]. It has achieved notable success across multiple domains, encompassing visual representation learning, pre-training of language models, and beyond [26]. Within the sphere of recommender systems, graph self-supervised learning (GSL) has gained prominence [2]. This paradigm has evolved from its initial unsupervised methods to sophisticated models that utilize network embeddings and random-walk proximity for self-supervision [27]. LightGCN, recognized for its simplicity and efficacy, has emerged as a favored graph self-supervised learning method, and forms the foundation of numerous graph contrastive learning (GCL) methods [9, 12]. Bayesian Personalized Ranking (BPR) loss, employed in LightGCN, is also a prevalent loss function in latest GCL methods [4]. However, within the context of complex networks, BPR loss relies on 2-hop path statistics for assessing sample relationships. Previous research has proved that relying solely on this local connectivity information may not suffice for accurately measuring the energy distance between nodes in the latent hyperbolic space. This limitation might neglect critical, broader network patterns, potentially limiting its effectiveness in biological analysis.

In this section, we develop a graph self-supervised learning framework using Topology-encoded Latent Hyperbolic Geometry for drug repositioning. By assigning ID embeddings to drugs and diseases and refining them across the drug-disease network, we aggregate these embeddings at various propagation layers to form the final network embedding. Multisource information about drugs and diseases is then integrated for drug repositioning tasks.

To summarize, this framework makes the following main contributions:

•We build TopoDR, a new computational drug repositioning framework based on lightGCN+, which integrates multimodal information.

•Empirical analyses on benchmark datasets reveal that our approach surpasses current drug repositioning algorithms in predictive accuracy.

### F.1 Materials and methods

In this section, we first delineate the benchmark datasets for drug repositioning, along with the multimodal representations related to drugs and diseases. Subsequently, we present the framework of the drug repositioning model, TopoDR, which primarily comprises two modules: the network embedding module and the prediction module. The primary contribution of this work lies in the network embedding module, developed through the graph self-supervised learning method, lightGCN+. This method is grounded in Topology-encoded Latent Hyperbolic Geometry. The prediction module integrates multimodal information to forecast alternative indications of drugs.

### F.1.1 Materials

The Fdataset and Cdataset are benchmark datasets in drug repositioning research. The Fdataset, recognized as the gold standard, comprises 1933 verified drug-disease associations, integrating 593 drugs from DrugBank and 313 diseases from the OMIM database [28, 29, 30, 31]. The Cdataset, in turn, widens the data collection spectrum, encompassing 663 drugs, 409 diseases, and 2352 drug-disease associations [32]. Together, these datasets offer a rich and diverse foundation of data, indispensable for the development and stringent validation of drug repositioning models.

Furthermore, we computed multimodal representations for drugs and diseases, informed by prior research [33]. For drugs, five types of similarities were calculated: chemical structure similarity $R_{cs}$ using SMILES and Tanimoto score [34]; ATC code similarity $R_{atc}$ based on therapeutic effects and chemical characteristics [35]; side effect similarity $R_{se}$ employing data from the SIDER database [36]; drug-drug interaction similarity $R_{ddi}$ derived from pharmacokinetics [29]; and target profile similarity $R_{dt}$ based on known drug targets [29]. For diseases, phenotype similarity $C_{ph}$ was assessed using phenotypic scores from the MimMiner database and disease ontology (DO) similarity $C_{do}$ based on gene ontology algorithms. We integrated these multimodal information sources by computing a weighted average of these similarity measures, described as follow:

$$R = \frac{R_{cs} + R_{atc} + R_{se} + R_{ddi} + R_{dt}}{5}. \tag{29}$$

$$C = \frac{C_{ph} + C_{do}}{2}. \tag{30}$$

### F.1.2 TopoDR for drug repositioning

TopoDR comprises three main components: a multimodal data integration component, a network embedding component, and a prediction component (Figure 8). We use ligthGCN+ as the network embedding component of the TopoDR framework. The prediction module utilizes both network embedding and multimodal data to depict drug-disease pairs, which can be defined as follows:

$$\text{Descriptor } (r, d) = [E_r, E_d, R_r, C_d] \tag{31}$$

We identify known associations as positive samples and correspondingly select an equal number of unacknowledged associations to serve as negative samples, thereby forming a training set. To predicting potential drug-disease associations, we employed a random forest algorithm as the classifier in this study.

### F.2 Experiments

### F.2.1 Performance evaluation

The 10-fold cross-validation was conducted to assess the performance of the drug repositioning model. In this process, all known drug-disease pairs were randomly divided into ten subsets, with each subset sequentially used as the test sample, while the remaining associations served as training samples. This cross-validation process was repeated ten times, and the averaged results were obtained. Additionally, six evaluation metrics were selected for assessing predictive performance: the area under the receiver operating characteristic curve (AUC), the area under the precision-recall curve (AUPR), accuracy (Acc.), precision (Pre.), recall (Rec.), and Matthew's correlation coefficient (MCC).

### F.2.2 Baseline model

To evaluate the performance of our proposed framework, we compared TopoDR with four existing drug repositioning methods listed below.

•deepDR [13] is a network-based deep learning model, integrating heterogeneous networks through a multi-modal deep autoencoder. It uses a variational autoencoder to encode and decode low-dimensional representations of drugs, identifying new uses for approved drugs.

•DRHGCN [14] is an approach using a graph convolutional network to integrate drug-drug, disease-disease similarities, and drug-disease associations. It combines intra-domain and inter-domain feature extraction with a layer attention mechanism for enhanced drug and disease embeddings.

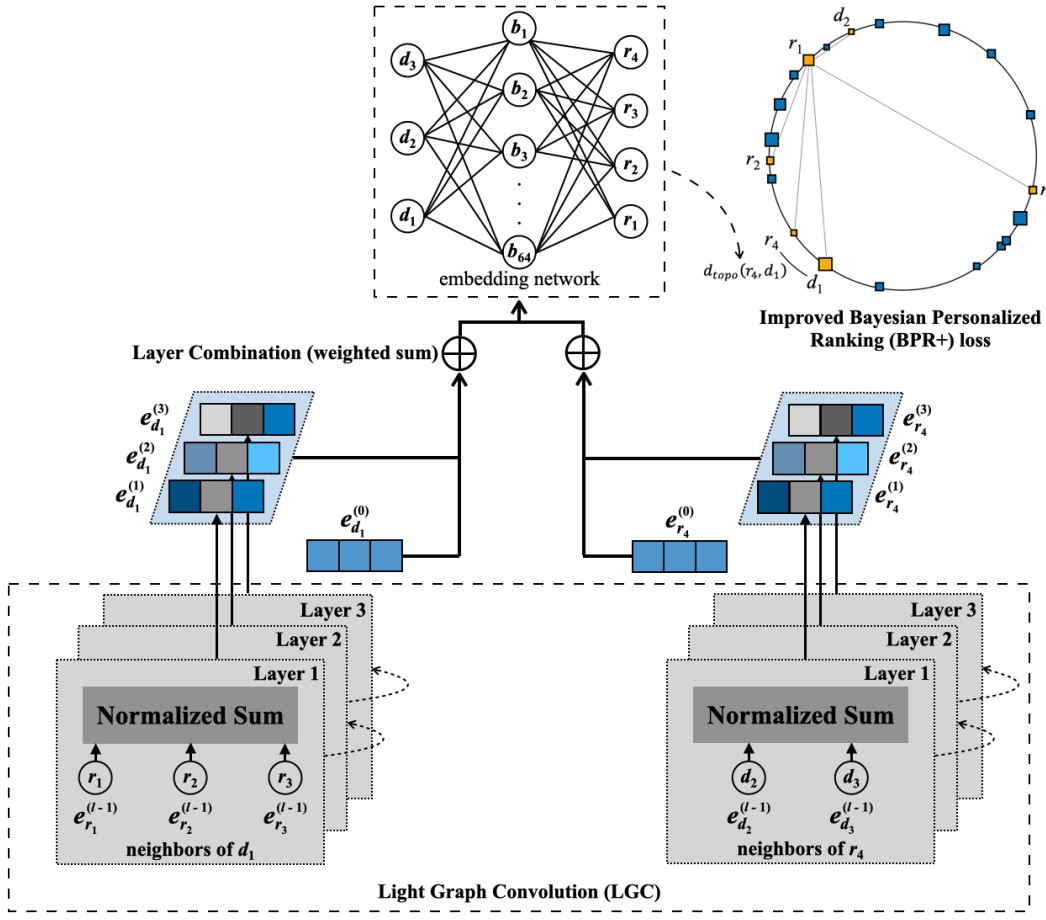

Figure 8: The flowchart of lightGCN+ model.

•DDAGDL[15] is a novel framework that integrates attention-based Graph Deep Learning with heterogeneous biomedical networks, enriched with biological knowledge. Utilizing non-Euclidean geometric priors and an attention mechanism, DDAGDL efficiently learns refined feature representations of drugs and diseases.

•AdaDR [16] is an adaptive GCN framework, leveraging an attention mechanism to fuse drug and disease features from topological and feature spaces for enhanced embeddings.

### F.2.3 The sensitivity analysis of parameters

In this section, we mainly examine the sensitivity analysis of parameters $\lambda$ and $\mu$ within 10 -fold cross-validation. TopoDR achieves its optimal performance when $\lambda$=1e-5 and $\mu$ = 1e-7. By altering one parameter while maintaining the other constant, we explore how such adjustments benefit the AUC values. It is noteworthy that we employed identical parameters for both the Fdataset and Cdataset.

Figure 9A depicts the performance trends of TopoDR with varying $\lambda$ values. As $\lambda$ varies, the AUC demonstrates a pattern of initial increase followed by a decrease. A downtrend in AUC is observed when $\lambda$ increases from 1e-5 to 1e-7, indicating the necessity of assigning higher weight to global topological structure similarity information. Similarly, Figure 9B presents the performance trends of TopoDR across different $\tau$ values, with the optimal AUC occurring at $\tau$=1e-7. A continuous decrease in AUC values is observed as $\tau$ is reduced from 1e-7 to 1e-9, suggesting that the regularization terms in Equation (7), specifically $\left\| E^0 \right\|^2$, should not be excessively reduced.

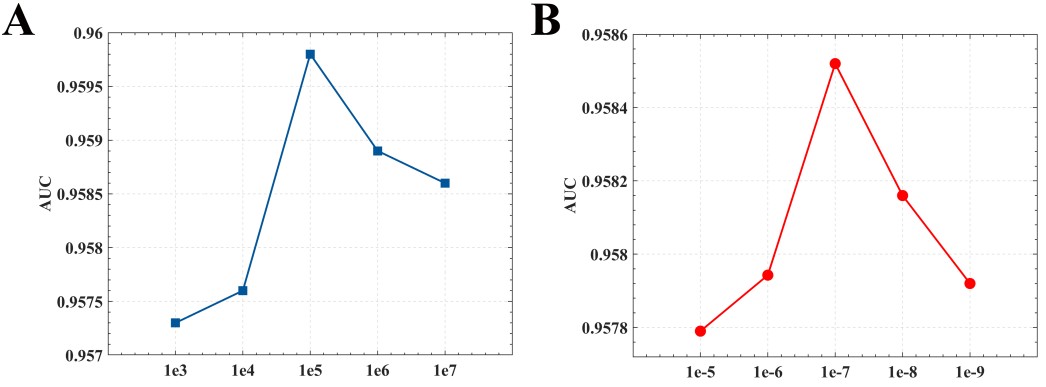

Figure 9: Effect of different $\lambda$ and $\tau$ on TopoDR performance. (A) Variation of the AUC values with the different settings of $\lambda$. (B) Variation of the AUC values with the different settings of $\tau$.

### F.2.4 Ablation study

In this section, the efficacy of TopoDR is evaluated through comparative training across datasets with diverse feature types. Two scenarios are analyzed, each predicated on the utilization of distinct modalities within the proposed framework:

•TopoDR-w/o-N: AdaDR without network information.

•TopoDR-w/o-A: AdaDR without attribute information.

Table 4 reports the performance of each module in TopoDR. It clearly demonstrates that both modalities of TopoDR enhance model performance, with the network information providing considerable predictive accuracy. TopoDR-w/o-A shows notable precision, achieving 0.9677 in the Fdataset and 0.9804 in the Cdataset, playing a crucial role in minimizing false positives. TopoDR-w/o-N exhibits a balanced performance across various metrics. In the Fdataset, it achieves an AUC of 0.9397, accuracy of 0.8471, and recall of 0.7454, whereas in the Cdataset, it scores an AUC of 0.9444, accuracy of 0.8544, and recall of 0.7555. These results indicate its strong overall predictive capability, enhancing the model's effectiveness in various aspects of prediction, from true positive rate to general accuracy and consistency. The synergistic interplay of these modalities enhances the model's robustness and adaptability across various predictive scenarios.

Table 4: Performance of models associated with different modality across all datasets.

| Dataset | Method | AUC | AUPR | Acc. | Pre. | Rec. | MCC |
|---------|--------|-----|------|------|------|------|-----|
| Fdataset | TopoDR-w/o-A | 0.9312 | 0.9453 | 0.7998 | 0.9677 | 0.6239 | 0.6430 |
| | TopoDR-w/o-N | 0.9397 | 0.9455 | 0.8471 | 0.9412 | 0.7454 | 0.7120 |
| | TopoDR | **0.9597** | **0.9581** | **0.8618** | **0.9683** | **0.7511** | **0.7440** |
| Cdataset | TopoDR-w/o-A | 0.9377 | 0.9529 | 0.8323 | **0.9804** | 0.6793 | 0.6989 |
| | TopoDR-w/o-N | 0.9444 | 0.9511 | 0.8544 | 0.9431 | **0.7555** | 0.7240 |
| | TopoDR | **0.9629** | **0.9686** | **0.8679** | 0.9753 | **0.7555** | **0.7556** |

### F.2.5 Comparison with graph self-supervised learning methods

To assess lightGCN+'s performance, we conducted comparisons with existing graph self-supervised learning methods (Node2vec [37], LINE [38], SDNE [39], and lightGCN [9]). As shown in Table 5, lightGCN+ consistently outperforms the other methods in almost all metrics, particularly in AUC and AUPR. These metrics are crucial as they represent the model's ability to distinguish between classes effectively. In the case of the Fdataset, lightGCN+ shows superior performance, leading in AUC and AUPR, and displaying high accuracy and MCC. These results indicate not only its effectiveness in classification tasks but also its reliability in handling imbalanced datasets, as suggested by the high MCC values. For the Cdataset, lightGCN+ again stands out, particularly in AUC, AUPR, and Precision, suggesting its strong ability to predict positive classes accurately while reducing false

positives. The slight trade-off between Recall and Precision in favor of the latter is noticeable for lightGCN+, suggesting a tendency towards conservative classification strategies that prioritize precision. Comparatively, other methods like Node2vec, LINE, SDNE, and lightGCN show varying degrees of effectiveness. Node2vec and lightGCN, for instance, exhibit balanced performance across most metrics, suggesting their versatility. However, SDNE tends to lag slightly, particularly in the Fdataset, indicating possible limitations in capturing complex graph structures or generalizing across different datasets.

In summary, strengths of lightGCN+ in handling graph-structured data with a strong emphasis on precision and overall classification effectiveness. Its consistent performance across different datasets underscores its robustness and potential for broader applicability in graph-related machine learning tasks.

Table 5: The performance of existing graph self-supervised learning methods on Fdataset and Cdataset.

| Dataset | Method | AUC | AUPR | Acc. | Pre. | Rec. | MCC |
|---------|--------|-----|------|------|------|------|-----|
| Fdataset | Node2vec | 0.9483 | 0.9472 | 0.8570 | **0.9698** | 0.7382 | 0.7360 |
|  | LINE | 0.9448 | 0.9399 | 0.8469 | 0.9514 | 0.7362 | 0.7144 |
|  | SDNE | 0.9256 | 0.9254 | 0.8244 | 0.9508 | 0.6875 | 0.6764 |
|  | lightGCN | 0.9540 | 0.9523 | 0.8495 | 0.9583 | 0.7336 | 0.7203 |
|  | lightGCN+ | **0.9597** | **0.9581** | **0.8618** | 0.9683 | **0.7511** | **0.7440** |
| Cdataset | Node2vec | 0.9537 | 0.9539 | 0.8685 | 0.9668 | 0.7650 | 0.7546 |
|  | LINE | 0.9516 | 0.9588 | **0.8687** | 0.9601 | **0.7713** | 0.7529 |
|  | SDNE | 0.9509 | 0.9542 | 0.8588 | 0.9711 | 0.7413 | 0.7395 |
|  | lightGCN | 0.9602 | 0.9592 | 0.8640 | 0.9655 | 0.7584 | 0.7469 |
|  | lightGCN+ | **0.9629** | **0.9686** | 0.8679 | **0.9753** | 0.7555 | **0.7556** |

### F.2.6 Case studies

To verify the practical applicability of TopoDR in identifying potential therapeutic drugs for complex diseases, we conducted case studies. Using TopoDR on Fdataset, we predicted unknown drug-disease associations. For each disease, candidate drugs are ranked in descending order based on their predictive scores. In recent years, the development of anti-tumor drug has received increasing attention [40]. In this context, we selected four common cancers (colorectal cancer, breast cancer, gastric cancer, and leukemia) for case studies and retrieved evidence for candidate drugs from the CTD database [41]. Table 6 lists the top 10 candidate drugs for these cancers as ranked by the TopoDR, with the confirmed therapeutic drugs from the CTD database highlighted in bold. In the United States, approximately 12% of women are estimated to develop breast cancer over their lifetimes, a statistic underscored by the recording of over 250,000 new cases in 2017 alone [42]. Our drug candidates have undergone preliminary biological validation for breast cancer treatment. For instance, the study by Buxant et al. investigates the impact of dexamethasone (Dex) on MCF-7 breast cancer cells. Their findings indicate that Dex can inhibit cell proliferation, potentially via a pro-apoptotic mechanism, suggesting its therapeutic potential [43]. Similarly, the emergence of cisplatin as a treatment for metastatic triple-negative breast cancer (mTNBC) is attributable to its role as a DNA-damaging agent [44]. The rationale for using cisplatin in mTNBC is derived from the subtype's unique molecular characteristics, notably its aberrant DNA repair mechanisms and extensive genomic instability. These features make mTNBC cells especially susceptible to DNA damage, thereby explaining the efficacy of cisplatin in this context. In summary, the outcomes of case studies indicate the practical significance of TopoDR in identifying potential therapeutic drugs for complex diseases.

Table 6: Top 10 candidate drugs for colorectal cancer, breast cancer, gastric cancer, and leukemia.

| Diseases (OMIM IDs) | Top 10 candidate drugs (DrugBank IDs) |
| --- | --- |
| Colorectal Cancer (D114500) | **Doxorubicin (DB00997)**; Teniposide (DB00444); **Prednisone (DB00635)**; **Cisplatin (DB00515)**; Bleomycin (DB00290); Busulfan (DB01008); Imatinib (DB00619); Conjugated Estrogens (DB00286); **Vincristine (DB00541)**; Zoledronic Acid (DB00399). |
| Breast Cancer (D114480) | Bleomycin (DB00290); **Cisplatin (DB00515)**; **Dacarbazine (DB00851)**; Salmon Calcitonin (DB00017); Methylprednisolone (DB00959); Teniposide (DB00444); **Vincristine (DB00541)**; Alendronic Acid (DB00630); Risedronic Acid (DB00884); Fludarabine (DB01073). |
| Gastric Cancer, Hereditary Diffuse (D137215) | Prednisone (DB00635); Vincristine (DB00541); Bleomycin (DB00290); Teniposide (DB00444); **Cisplatin (DB00515)**; **Imatinib (DB00619)**; Dacarbazine (DB00851); Zoledronic Acid (DB00399); Methylprednisolone (DB00959); Azathioprine (DB00993). |
| Breast Cancer (D109543) | **Dexamethasone (DB01234)**; **Prednisone** ; **Cisplatin (DB00515)**; **Doxorubicin (DB00997)**; Triamcinolone (DB00620); **Daunorubicin (DB00694)**; **Zoledronic Acid (DB00399)**; Vinblastine (DB00570); Pamidronic Acid (DB00282); **Methylprednisolone (DB00959)**. |

