# OpenReview forum: "Graph-Theoretic Insights into Bayesian Personalized Ranking for Recommendation"
_NeurIPS.cc/2025/Conference — NeurIPS 2025 poster_

### Official Review · Reviewer_uD2G · 2025-06-25

**Clarity:** 2
**Significance:** 2
**Originality:** 3
**Rating:** 4
**Confidence:** 4

**Summary:**

This paper analyzes the limitations of BPR loss in graph self-supervised learning (GSL) from a graph-theoretic perspective. The author argues that BPR relies solely on 2-hop paths for node-relation evaluation, ignoring global topology. They propose BPR+, a novel loss function incorporating even-hop paths via TopoLa distance (derived from latent hyperbolic geometry) to capture global connectivity.

**Questions:**

See the Weaknesses

**Ethical Concerns:**

["NO or VERY MINOR ethics concerns only"]

**Final Justification:**

After carefully considering the authors’ rebuttal and the other reviewers’ comments, I have decided to keep my original score unchanged. The authors have satisfactorily addressed most of my specific concerns—especially regarding missing baselines and clarifications on experimental settings. However, the overall contribution to the broader field remains incremental; the proposed method offers only modest improvements over existing approaches and does not substantially advance the community’s understanding of the problem. Consequently, I maintain my initial rating.

**Limitations:**

Yes

**Quality:**

3

**Strengths And Weaknesses:**

Strengths

1. This paper is solid, and most of the theories are accompanied by corresponding theoretical analyses.

2. This paper conduct complete experimental verification to demonstrate the rationality of its analyses and methods, and the proposed BPR+ has shown promising performance improvements.

Weaknesses

1. The inclusion and meaning of the abstract node need to be more clearly presented, since this is the reason why the calculation method of the original BPR loss can be understood as a 2-hop path. This is also one of the core issues addressed in this paper.

2. Although the paper provides reasonable insights, the novelty of the proposed BPR+ has limitations.

3. There is a lack of comparison with related work. More relevant work would help readers understand the background and the contributions of this paper.

---

> ### Author Rebuttal · Authors · 2025-07-29
>
> We thank reviewer uD2G for the concise summary of our paper. Below we address his/her concerns.
>
> **Question 1:**
> > "The inclusion and meaning of the abstract node need to be more clearly presented, since this is the reason why the calculation method of the original BPR loss can be understood as a 2-hop path. This is also one of the core issues addressed in this paper."
>
> **Response:**
> We are appreciative to the reviewer for the thoughtful suggestion. The concept of "abstract node" is used as a theoretical tool to reinterpret the inner product in BPR from a graph perspective. Specifically, the user embedding $\mathbf{e}_u$ and item embedding $\mathbf{e}_v$ are both vectors in the same latent space, and their inner product $\langle \mathbf{e}_u, \mathbf{e}_v \rangle$ can be viewed as a projection through a shared set of latent dimensions. Each dimension of this space can be interpreted as an abstract node that connects to both users and items.
>
> To make this more interpretable in topological terms, we consider each latent dimension as an "abstract node" that connects to all users and items. Under this view, the inner product corresponds to counting 2-hop connections: $user \rightarrow abstract node \rightarrow item$.
>
> This analogy allows us to model the BPR loss as a function of 2-hop connectivity in a user-item-abstract network. This formulation reveals that BPR only captures coarse structural similarity (i.e., 2-hop paths) and motivates us to extend it to TopoLa distance, which incorporates higher-order even-hop paths to better reflect global graph topology.
>
> **Question 2:**
> > "Although the paper provides reasonable insights, the novelty of the proposed BPR+ has limitations."
>
> **Response:**
> We are grateful to the reviewer for the thoughtful comments on our method. The primary novelty of our work lies in re-examining the widely used BPR loss from a graph-theoretic and network geometry perspective. To the best of our knowledge, this line of thinking has not been explored before. This reinterpretation not only reveals the limitations of the conventional inner product formulation, which effectively corresponds to 2-hop connectivity, but also provides a foundation for integrating latent hyperbolic geometry with graph neural networks.
>
> Based on this theoretical understanding, we introduce the TopoLa distance as a fine-grained topological metric derived from multi-hop path structures, going far beyond BPR's coarse similarity modeling. Building upon this insight, we propose BPR+, which can be viewed as a significantly generalized framework. Its relevance extends beyond BPR, as many other widely-used techniques in graph representation learning, including InfoNCE, self-attention, and contrastive learning, also rely on inner product similarity. Our work highlights the limitations of such similarity measures in theory and provides a new direction for rethinking them from a topological perspective.
>
> We believe this shift offers valuable insight for the community, and hope that future research will further build upon this line of thought.
>
> **Question 3:**
> > "There is a lack of comparison with related work. More relevant work would help readers understand the background and the contributions of this paper."
>
> **Response:**
> We sincerely thank the reviewer for this helpful suggestion. We review many representative extensions of BPR. Most of them focus on improving input features, loss functions, or specific application scenarios, while keeping the core scoring function, namely the inner product, unchanged. For example, ABPR [1] models feedback uncertainty, VBPR [2] integrates visual features, DPR [3] addresses item recommendation bias, and LkP [4] focuses on ranking diversity. These methods significantly improve BPR from different perspectives, but none of them question or modify the use of the inner product itself.
>
> In contrast, our work focuses on this fundamental assumption. We revisit BPR from a geometric and graph-theoretic perspective and analyze how the inner product may fail to capture meaningful topological relationships, especially under conditions such as over-smoothing. Based on this observation, we propose using a network geometry-based distance (TopoLa) to replace the inner product as a new way to compute sample similarity.
>
>
> [1] Pan W, Zhong H, Xu C, et al. Adaptive Bayesian personalized ranking for heterogeneous implicit feedbacks[J]. Knowledge-Based Systems, 2015, 73: 173-180.
> [2] He R, McAuley J. VBPR: visual bayesian personalized ranking from implicit feedback[C]//Proceedings of the AAAI conference on artificial intelligence. 2016, 30(1).
> [3] Zhu Z, Wang J, Caverlee J. Measuring and mitigating item under-recommendation bias in personalized ranking systems[C]//Proceedings of the 43rd international ACM SIGIR conference on research and development in information retrieval. 2020: 449-458.
> [4] Liu Y, Walder C, Xie L. Learning k-determinantal point processes for personalized ranking[C]//2024 IEEE 40th International Conference on Data Engineering (ICDE). IEEE, 2024: 1036-1049.

---

> > ### Comment · Reviewer_uD2G · 2025-08-01
> >
> > After careful consideration, we decide to maintain the score.
> > Thanks for the author‘s response, the response has solved our questions about the technique details to some extent.

---

> > > ### Author Response · Authors · 2025-08-04
> > >
> > > Dear Reviewer uD2G, thank for your positive feedback and your valuable time in evaluating our work. We are glad that our rebuttal has solved your questions to some extent. We will be happy to clarify any further questions you may have.

---

### Official Review · Reviewer_HFzz · 2025-06-30

**Clarity:** 2
**Significance:** 3
**Originality:** 3
**Rating:** 5
**Confidence:** 4

**Summary:**

The paper re-interprets the common BPR loss function in recommendation under the lens of hyperbolic geometry to work on its current limitations and improve them.

Concretely, the authors start by observing that the calculation of the predicted user-item interaction score, namely can be also re-interpreted as the $(u, i)$ entry of the Gram matrix of the node embeddings. This allows to go from the original user-item graph topology to another network involving user and item nodes, along with some dummy nodes representing the embedding latent elements. Thus, calculating $\hat{y}_{ui}$ eventually implies assessing the number of 2-hops connecting any user and item in the graph through the dummy embedding node. While the number of 2-hops connections is positively correlated to the energy distance between nodes in the hyperbolic space, the authors claim only using 2-hops distances is not sufficient.

Thus, they propose an improved version of the BPR loss, dubbed BPR+, where the computation of $\hat{y}_{ui}$ is replaced with the calculation of the topological distance between the two nodes, computed at even-numbered hops (i.e., 2-4-6-... hops). This improved loss has the advantage of incorporating more global information and capturing the topological structure between users and items in the system.

Finally, as the computation of the topological distances is computationally expensive (because it involves a matrix power repeated multiple times), the authors propose to work on the decomposed representation of node embeddings through SVD.

Extensive experiments on five recommendation datasets, when testing the BPR+ loss on five baselines, demonstrate the improved recommendation performance with respect to the shallow models. Further analyses considering the relation between the topological similarity and the loss, and the union of neighbors and loss, provide another important perspective on the proposed loss. Moreover, an efficiency study shows how the integration of the SVD decomposition can greatly improve the training time of the model. Finally, the authors efficiently apply the same proposed methodology to another similar domain and task, namely, drug repositioning, demonstrating how BPR+ can outperform other baselines on this task.

**Questions:**

My questions are mainly referred to the last three weaknesses I raised.

Q1) To which power the computations in equations 10-11 are computed?

Q2) Can the proposed approach be potentially applied to monopartite graphs (i.e., graphs with nodes from one single partition, such as citation networks)? And, in case, which would it be the implication of that? Could it be beneficial also to them?

Q3) The literature has shown how computing more than 1 negative for each positive item in the BPR loss can be beneficial to improve the performance [\*]. How this would impact on your loss function? Would this increase the computational complexity?

**References**

[\*] Steffen Rendle, Walid Krichene, Li Zhang, and John R. Anderson. 2020. Neural Collaborative Filtering vs. Matrix Factorization Revisited. In RecSys. ACM, 240–248.

**Ethical Concerns:**

["NO or VERY MINOR ethics concerns only"]

**Final Justification:**

After the rebuttal phase, after reading the other reviews, and authors' responses to those, I decide to rise my initial score I gave to the paper to "Accept".

Specifically, all my concerns have been discussed and addressed by the authors.

Whatever the final outcome for this paper, I think it will benefit quite a lot from the fruitful discussion we had along with the authors, the reviewers, and the Area Chair. I believe the paper will improve its quality a lot in any case.

**Limitations:**

Yes.

**Paper Formatting Concerns:**

To the best of my understanding, I do not notice any major formatting issue in the paper.

**Quality:**

3

**Strengths And Weaknesses:**

**Strengths**

$\bullet$ The paper has the merit to question the goodness of one of the most established loss methodologies in the literature, namely, BPR, in the realm of graph self-supervised learning. Thus, it can hold great significance in the current literature on the topic.

$\bullet$ The proposed methodology seems adequately sound, being supported by theoretical proofs for each introduced theorem (as also evidenced in the Supplementary Material part).

$\bullet$ It is nice to see how the proposed approach can be applied to any graph-based recommender system without necessarily modifying its structure and methodology underneath.

$\bullet$ The experimental section is quite clear and shows the efficacy of the model under different perspectives.

$\bullet$ Remarkably, the authors apply the proposed approach for another task and domain, namely, drug repositioning, demonstrating how the approach can go further the scenario of personalized item recommendation.

**Weaknesses**

$\bullet$ To the best of my understanding, the code is not released at review time, thus not ensuring the complete reproducibility of the work. While the authors acknowledge the usage of the framework SSLRec, I think it would have been beneficial for them to provide the code for their implemented work.

$\bullet$ To the best of my understanding, it is not clear to which power the matrix computation (equations 10 and 11) is actually performed. The authors should have better clarified this aspect.

$\bullet$ I am wondering which would be the implications of applying the proposed strategy to monopartite graphs (instead of bipartite graphs as in the paper's case).

$\bullet$ Moreover, I am also wondering how the computational complexity of the model would change if, in the BPR schema, we sample more than 1 negative item per each positive item. The literature has shown how this could potentially improve results, and I think that analyzing this aspect would be beneficial to improve the quality of the work.

---

> ### Author Rebuttal · Authors · 2025-07-29
>
> We are grateful for the reviewer's thoughtful engagement with our work and their constructive suggestions. Below are our responses to each raised point.
>
> **Question 1:**
> > "To which power the computations in equations 10-11 are computed?"
>
> **Response:**
> We thank the reviewer for raising this valuable point regarding the efficiency of computing TopoLa distance. In our formulation, Eq.10 represents a direct matrix-based calculation of even-hop path counts between nodes. It is designed primarily to intuitively illustrate how structural information is aggregated across multiple hops. While it captures all even-length paths, repeated matrix multiplications (e.g., $\boldsymbol{A}^2$, $\boldsymbol{A}^4$, ..., $\boldsymbol{A}^{2k}$) can indeed lead to high computational cost, especially for large $k$. This limitation is also reflected in the runtime analysis shown in Figure 5.
>
> To address this, we introduce Eq.11, which is derived from Eq.10 via singular value decomposition (SVD). By applying SVD to the embedding matrix, the even-hop contributions are encoded into the diagonal matrix $\boldsymbol{\Sigma}$, and since only its diagonal elements are non-zero, the cost of computing high powers of $\boldsymbol{\Sigma}$ becomes constant-time, i.e., $O(1)$ for each exponentiation. This dramatically reduces the overall computational burden of accumulating even-hop contributions.
>
> Because of this optimization, the cost of computing even-hop path contributions becomes independent of the hop count, allowing us to incorporate an arbitrary number of hops without any significant increase in runtime. In our experiments, we observed that contributions from paths beyond 40 hops have a negligible effect on the final TopoLa distance. Therefore, we truncate the expansion at 40 hops to balance accuracy and efficiency.
>
> **Question 2:**
> > "Can the proposed approach be potentially applied to monopartite graphs (i.e., graphs with nodes from one single partition, such as citation networks)? And, in case, which would it be the implication of that? Could it be beneficial also to them?"
>
> **Response:**
> Thanks for raising this subtle question. Our method is inherently compatible with monopartite graphs. As shown in the definition of the embedding matrix $\boldsymbol{E}$ in the paper, we concatenate user and item embeddings into a single matrix. This formulation treats users and items as nodes of the same type within a unified latent space, which aligns well with the concept of a monopartite graph.
>
> Moreover, the proposed TopoLa distance is defined based on the connectivity patterns (i.e., multi-hop paths) between nodes, regardless of their original semantic roles or partitions. Whether the graph consists of user-item pairs or homogeneous nodes (e.g., in citation networks), our method captures the global structural relationships through even-hop connectivity and thus remains applicable and effective in such scenarios.
>
> **Question 3:**
> > "The literature has shown how computing more than 1 negative for each positive item in the BPR loss can be beneficial to improve the performance [*]. How this would impact on your loss function? Would this increase the computational complexity?"
>
> **Response:**
> We thank the reviewer for asking this interesting question. Our current implementation of BPR+ follows the original BPR setting, which samples one negative item per positive instance. However, as the reviewer observed, sampling multiple negatives per positive can often lead to improved model performance.
>
> The proposed BPR+ framework can naturally accommodate this extension. In fact, the TopoLa-based formulation remains unchanged under multiple negative sampling, as the abstract network is constructed per batch and the TopoLa distance is computed based on all sampled pairs. The only difference lies in the increased number of user-item pairs within each batch.
>
> This extension would linearly increase the computational cost with respect to the number of negatives per positive, but the additional overhead remains tractable, especially given the fact that our method uses an SVD-based optimization in TopoLa computation. We appreciate the suggestion and will consider including this variant in future work.

---

> > ### Comment · Reviewer_HFzz · 2025-08-01
> >
> > Dear Authors,
> >
> > Thanks for your time in answering to my raised weaknesses and questions.
> >
> > Thank you for the clarification on the computation power in your methodology. It is quite remarkable you could reach 40 hops and still finding a good balance between performance and efficiency.
> >
> > Thank you also for clarifying the aspect regarding monopartite and bipartite graphs. Indeed, I now understand your approach is not specifically considering the semantics behind the graph edges, but rather the difference between nearest neighbors and even-hop neighbors.
> >
> > Finally, I agree the possible integration of BPR with multiple sample negatives could represent an important and interesting avenue for future directions of your work.
> >
> > I do not have any other concerns regarding the paper. The answers you gave helped confirming the initial positive judgment I had on the paper, and clarified some still-obscure points I had raised. Thus, I will increase my score to "Accept", which I believe is representative of the quality of your submission.

---

> > > ### Author Response · Authors · 2025-08-04
> > >
> > > Dear Reviewer HFzz， thank you very much for your positive feedback and favorable consideration of our work! We are glad that our rebuttal addressed your concerns, and we are happy to clarify any further questions you may have.

---

### Official Review · Reviewer_fhF3 · 2025-07-01

**Clarity:** 3
**Significance:** 3
**Originality:** 3
**Rating:** 5
**Confidence:** 3

**Summary:**

This paper focuses on the field of graph self-supervised learning, analyzing the widely used but rarely examined BPR (Bayesian Personalized Ranking) loss function. It identifies and verifies its weaknesses in capturing global connectivity and topological similarity from multiple perspectives. Building on the perspective of latent hyperbolic geometry, the authors propose a more effective distance metric called TopoLa and its corresponding loss function BPR+. The improvements brought by BPR+ are validated on five public datasets, enhancing the performance of LightGCN and other contrastive learning models. Furthermore, in the field of drug repositioning, the proposed model achieves significant improvements over existing approaches.
The main contributions of this paper are as follows:
1.	Revealing the limitations of traditional BPR loss.
2.	Analyzing the effectiveness of common graph convolution operations under the latent hyperbolic framework.
3.	Proposing a novel loss function, BPR+, which enhances model performance.
4.	Optimizing the computational complexity of BPR+.
5.	Conducting extensive experiments to demonstrate the superiority of proposed method, establishing a new framework for drug repositioning.

**Questions:**

Firstly, the paper acknowledges the over-smoothing phenomenon in higher layers of LightGCN while proposing the new method involving multi-layer aggregation. I would appreciate clarification on whether the TopoLa-based approach is susceptible to over-smoothing, or if specific design optimizations have been implemented to prevent this issue.
Then, the proposed methodology demonstrates that the addition/removal of a single node can influence representation learning across the entire graph. This characteristic raises concerns about its adaptability to real-world scenarios with dynamic user/item pools. I would like to understand the authors' analysis and potential improvements regarding this aspect.
Third, beyond time complexity, space complexity represents another critical consideration for graph algorithms, as storing intermediate node states often requires substantial memory resources. Has the new method been evaluated in terms of its space complexity requirements?
These three points constitute my primary questions after careful reading. Satisfactory explanations to these concerns could lead to an 'Accept' rating.

**Ethical Concerns:**

["NO or VERY MINOR ethics concerns only"]

**Final Justification:**

I confirm that all my questions have now been satisfactorily addressed.

**Limitations:**

Yes, the authors have addressed the limitations in their paper, and I have provided further suggestions in the previous part.

**Quality:**

3

**Strengths And Weaknesses:**

First of all, regarding the quality of proposed TopoLa technique, its derived methods BPR+ and TopoDR demonstrate statistically significant superior performance over existing models in experiments, confirming the effectiveness of the approach. Furthermore, this work provides comprehensive theoretical foundations for the technique's validity from the perspective of latent hyperbolic geometry, including mathematical proofs and physical property analysis, making it a thorough and well-substantiated research contribution.
Secondly, regarding the clarity, this paper begins with a systematic analysis of BPR's limitations, employs latent hyperbolic geometry as a robust analytical framework, and subsequently develops novel methods with computational complexity optimizations. The work culminates in comprehensive experimental validation of the proposed approach's effectiveness. The logical progression is complete and well-structured, with particularly clear presentation. Furthermore, the inclusion of source code and datasets in the appendix enhances reproducibility, enabling verification of the reported results.
Then, regarding the significance of this work, the proposed novel metric represents a breakthrough in traditional loss functions through cross-domain knowledge integration by the research team. Notably, it exhibits considerable model-agnostic characteristics, enabling broad applicability across various graph-structured algorithms. These features suggest substantial potential for future utilization and development in the field, but it still potentially possesses a bit of complexity issue which will be addressed in the Question part.
Finally, the proposed even-hop paths based method in this paper can be regarded as an extension of traditional BPR loss that incorporates insights from latent hyperbolic geometry. By condensing multi-level topological structure information into multi-hop path representations, this approach achieves more accurate capture of topological similarity and global connectivity. Thus, the underlying originality and reasoning is well-justified.

---

> ### Author Rebuttal · Authors · 2025-07-29
>
> We thank reviewer fhF3 for his/her thorough and insightful comments on our method as well as its theoretical analysis. We are highly appreciative of the positive rating. Below are our responses to each raised point.
>
> **Question 1:**
> > "Firstly, the paper acknowledges the over-smoothing phenomenon in higher layers of LightGCN while proposing the new method involving multi-layer aggregation. I would appreciate clarification on whether the TopoLa-based approach is susceptible to over-smoothing, or if specific design optimizations have been implemented to prevent this issue."
>
> **Response:**
>   We are grateful to the reviewer for raising this subtle and important issue. Compared to BPR, the introduction of latent hyperbolic geometry in BPR+ alleviates the negative effect of over-smoothing on relationship estimation.
>
>   Over-smoothing refers to the convergence of node features as the network depth increases, leading to indistinguishable embeddings across nodes. In the case of BPR, which relies on inner product to evaluate pairwise relationships, this convergence causes the scores between different sample pairs to become numerically similar. As illustrated in Figure 4a, when dot products are concentrated around a constant value (e.g., 20), the corresponding topological similarities can still vary significantly (e.g., from 20% to 50%). This implies that BPR could yield coarse-grained and uninformative relationship scores under over-smoothed conditions.
>
>   Contrarily, BPR+, by incorporating latent hyperbolic geometry, is more sensitive to subtle structural differences in the graph. Even if node features are similar, the TopoLa distance is still able to reflect the variation in topological similarity between nodes (Figure 4b). As a result, BPR+ provides a more fine-grained and discriminative measurement of node relationships, even in the presence of over-smoothing.
>
> **Question 2:**
> > "Then, the proposed methodology demonstrates that the addition/removal of a single node can influence representation learning across the entire graph. This characteristic raises concerns about its adaptability to real-world scenarios with dynamic user/item pools. I would like to understand the authors' analysis and potential improvements regarding this aspect."
>
> **Response:**
>   We sincerely thank the reviewer for this insightful and important question. We acknowledge that the TopoLa distance is conceptually defined over the entire abstract network and thus reflects a form of global structural modeling. However, in practice, our method does not compute TopoLa distance over the full user-item graph at each step. Instead, the BPR+ loss is calculated within each mini-batch, and the corresponding abstract network is constructed only from the users and sampled items in that batch. This means that the actual computations are localized, and the full graph structure is not directly involved in each training step.
>
>   Taking LightGCN as an example, with a batch size of 1,500 and a network containing 40,000 items, introducing or removing a single item changes its sampling probability by approximately $1500/40000 = 3.75\%$. This change is minor and only affects whether the node is sampled into a future batch. As a result, the composition of nodes in any specific batch remains relatively stable, and the structure of the corresponding abstract network exhibits minimal variation.
>
>   Therefore, while the TopoLa distance can reflect how structural relations change with node addition/removal, such changes only take effect when the node is actually sampled into the batch. Since loss computation is restricted to the batch, the representation learning process remains stable and unaffected by a single node's presence or absence unless it is sampled.
>
>   In summary, the proposed method benefits from structural awareness via TopoLa, but the training is performed on localized, batch-level networks, which makes it naturally robust to small-scale perturbations in the overall user/item pool.
>
> **Question 3:**
> > "Third, beyond time complexity, space complexity represents another critical consideration for graph algorithms, as storing intermediate node states often requires substantial memory resources. Has the new method been evaluated in terms of its space complexity requirements? These three points constitute my primary questions after careful reading."
>
> **Response:**
>   We are very thankful for the constructive comments and suggestions. As the main modification of BPR+ lies in replacing the inner product with the TopoLa distance, the space complexity primarily depends on how the relevant matrix representations are stored and computed.
>
>   In BPR, computing the inner product requires storing the embedding matrix $\boldsymbol{E} \in \mathbb{R}^{(m+n) \times N_e}$, where $m$ and $n$ are the numbers of items and users, respectively, and $N_e$ is the embedding size. This leads to a space complexity of $O((m+n) N_e)$.
>
>   In contrast, BPR+ introduces an additional step of SVD decomposition on $\boldsymbol{E}$, producing three matrices:   $\boldsymbol{U} \in \mathbb{R}^{(m+n) \times r}$, $\boldsymbol{\Sigma} \in \mathbb{R}^{r \times r}$, and  $\boldsymbol{V} \in \mathbb{R}^{r \times N_e}$, where $r$ is the rank of $\boldsymbol{E}$ satisfying $r < \min(m+n, N_e)$. Therefore, the space complexity for computing the TopoLa distance is $O((m+n)r + r^2)$. Since $r$ is typically smaller than $N_e$, the overall space cost of BPR+ remains manageable and, in some cases, even lower than that of directly storing the full matrix $\boldsymbol{E}$.
>
> To provide a concrete example:
>
>   In our experiments with LightGCN, each batch typically contains $m = 1500$ users and $n = 3000$ items, with embedding size $N_e = 64$. The SVD rank $r$ is also set to $64$ for simplicity.
>
> - **The space cost of BPR is**:
>   $(m+n) \times N_e = 4500 \times 64 = 288000$
>
> - **The space cost of BPR+ is**:
>   $(m+n) \times r + r^2 = 4500 \times 64 + 64 \times 64 = 292096$
>
>
>
> Thus, BPR+ introduces only an additional 4,096 units of memory, corresponding to a 1.4% increase in space complexity compared to BPR.
>
>   Considering the performance improvements gained through TopoLa and the small overhead incurred, we believe this trade-off is acceptable and practical.

---

> > ### Comment · Area_Chair_1P47 · 2025-08-06
> > **Your response**
> >
> > Dear Reviewer fhF3,
> > can you kindly go over the rebuttal to see if your questions and concerns are answered. Please respond to the reviewers. sincerely AC

---

> > ### Comment · Reviewer_fhF3 · 2025-08-07
> >
> > Thanks for your comprehensive responses to my concerns.
> >
> > First, regarding the potential impacts of over-smoothing, I appreciate your explanation of how the phenomenon takes place and how the BPR+ method prevents excessive topological gaps between nodes with similar features. This clarification has significantly alleviated my initial concerns.
> >
> > Second, concerning robustness in variable pool, the quantitative analysis demonstrating limited practical impact of global user/item pool variations on training is particularly valuable. This evidence substantially strengthens the model's practical applicability.
> >
> > Third, for model complexity,  I equally appreciate the quantitative approach that clearly illustrates your model's complexity characteristics. While there are some memory storage requirements, they are clearly justified by the performance improvements - a pragmatic trade-off that reflects the work's solid engineering foundations.
> >
> > I confirm that all my questions have now been satisfactorily addressed. Accordingly, I am revising my rating to Accept. However, I would recommend incorporating these clarifications more explicitly in the manuscript, as they would greatly help readers understand:
> > 1. The model's inherent strengths and limitations
> > 2. Its practical deployment considerations
> > 3. The technical rationale behind key design choices
> > This additional transparency would further enhance the paper's educational value for researchers facing similar challenges.

---

> > > ### Author Response · Authors · 2025-08-07
> > >
> > > Dear Reviewer fhF3, thank you so much for your valuable time and effort in evaluating our paper.  We really appreciate your positive feedbacks and willingness in raising the score. We are glad that our rebuttal has addressed all your concerns, and will follow your suggestions when preparing for the final version.

---

### Decision · Program_Chairs · 2025-09-17

**Decision:**

Accept (poster)

**Comment:**

This paper furthers the understanding of Bayesian Personalized ranking and develops new algorithms.
At the heart of the new results is a new measure of Topological distance through the use of hyperbolic geometry. There was overall appreciation of the work. Concerns raised by the reviewers was addressed by the authors during the rebuttal. One of the key concerns was the comparison with other global strategies such as Topola. Authors pointed out that the proposed algorithm was different than Topola. There was consensus that this paper may be considered as accepted.
However during the discussion phase no reviewers championed this paper.
Overall assessment is this paper could be of interest to graph neural networks community attending to Neurips community.